# The impact of COVID-19 pandemic on fertility behaviour in Indian states: Evidence from the National Family Health Survey (2019/21)

Md. Mahfuzur Rahman[1]*, Manas Ranjan Pradhan[2], Manoj Kumer Ghosh[3], Md. Moshfiqur Rahman[4]

1 Department of Population Science and Human Resource Development, University of Rajshahi, Rajshahi, Bangladesh, 2 Department of Fertility and Social Demography, International Institute for Population Sciences (IIPS), Mumbai, India, 3 Department of Geography and Environmental Studies, University of Rajshahi, Rajshahi, Bangladesh, 4 Department of Public Health and Hospital Administration, National Institute of Preventive and Social Medicine (NIPSOM), Dhaka, Bangladesh

* mahfuz_pops@ru.ac.bd

**Data Availability Statement:** The data can be accessed for free from the official open-access repository of the Demographic and Health Survey (DHS) program (https://dhsprogram.com/data/

## Abstract

### Background

The COVID-19 pandemic affected a broad spectrum of people's lives very quickly. Although the pandemic could influence people's fertility behaviours in several ways, there is little knowledge about such influence in diverse socioeconomic and cultural settings. This study investigated the impact of the COVID-19 pandemic on fertility behaviours and desires among women in a lower-middle-income country, India.

### Data and methods

Our study analyzed cross-sectional data from 13 states and union territories (UTs) in India that were surveyed in pre- and post-lockdown periods by the 2019/21 National Family Health Survey (NFHS). The data were analysed using the descriptive analysis technique and the multilevel logit model. All these analyses were performed using the technique developed for complex sample design.

### Results

The poverty-stricken states of Uttar Pradesh and Jharkhand and the Odisha state with moderate socioeconomic status experienced a significant decrease in contraceptive use and non-significant changes in the desire for birth and sexual activities after the lockdown. Contraceptive use significantly increased after the lockdown in the rich states of Punjab, Delhi, and Tamil Nadu, as well as in the Arunachal Pradesh state with moderate socioeconomic status. The changes in fertility behaviours in Uttar Pradesh, Jharkhand, and Odisha may influence fertility positively, while those changes in Punjab, Delhi, Tamil Nadu, and Arunachal Pradesh may influence fertility negatively.

dataset/India_Standard-DHS_2020.cfm?flag=1).
The DHS data can be downloaded for free by
registering for data and requesting data for use.
The authors confirm that others may download the
data in the same way the authors did, and the
authors did not receive any special advantage in
accessing the data. The title of the data files used
for this study were IAIR7ESV and IAIR7EDT from
the 2019/21 DHS survey in India.

**Funding:** The author(s) received no specific
funding for this work.

**Competing interests:** The authors have declared
that no competing interests exist.

## Conclusion

At the aggregate level, there was a significant increase in desire for another child and a
decrease in contraceptive use after the lockdown, which may influence fertility positively but
can be compensated by reduced sexual activities. The influence of the fertility trends in the
states with pro-natalist changes on India's recent fertility trend could be greater than those
with anti-natalist changes, which can be better understood by analyzing reliable data from a
couple of years following the 2019/21 NFHS.

## Introduction

The coronavirus disease 2019 (COVID-19) has significantly impacted all people's lives in a
number of ways. The pandemic forced countries to go into complete lockdown in 2020, which
lasted several weeks or months [1]. The first confirmed case of COVID-19 in India was
reported on January 30, 2020 [2]. The rapid spread of COVID-19 forced the Indian govern-
ment to impose lockdown throughout the country to control the pandemic. The COVID-19
pandemic affected lifestyles and economic activities in India at a massive level [2–4]. The
changes in lifestyles, livelihoods, and health care systems caused by the COVID-19 pandemic
and associated lockdown are expected to reshape the fertility behaviours of people [5].
Although the pandemic may affect human fertility in several ways, the effect of social alien-
ation, in particular, lockdown-type measures, remains enormous [6]. The lockdown may
cause delayed marriages, detachments of the couples, deterioration of family relationships,
economic and health uncertainty, and increased workload for women due to the closure of the
institutions [7–12]. All these factors may negatively affect fertility. On the contrary, staying
home and solitude might offer the chances for spending more time with one's partner and
enhancing the relationship quality, which in turn might influence couples to enlarge their fam-
ily sizes [13–15]. The lockdown may also result in limited access to family planning services
and a disruption to service deliveries [5, 16, 17]. A disruption to the use of family planning ser-
vices is also evident in India, as reflected in the lower use of such services after the onset of the
pandemic and during the lockdown period [18, 19]. All these factors may exert positive effects
on fertility.

Currently, 18% of the world's population resides in India, a lower-middle income country
in South Asia [20, 21]. India is expected to outrank China in 2023 as the most populous coun-
try in the world [22]. The noticeable decreases in the total fertility rates (TFR: the average
number of births per woman) in India and in all of its states since 1992/93 allowed their
administrations to effectively control their rapid population growths [23, 24]. The decreases in
fertility rates in India and its states were driven by the apparent changes in fertility behaviours
of their people, which primarily include the decline in the mean ideal family size [24]. The
decline in mean ideal family size in India were driven by modernization resulting from wom-
en's education, paid employment for women, improving the standard of living, urbanization,
regular exposure to the media, and increased child survival [25, 26]. Family planning pro-
gramme activities (such as spreading family planning messages through mass media and dis-
tributing contraception through public sectors and non-government organizations) and the
declined ideal family size in turn led to a noticeable increase in the use of contraception in
India and almost all of its states between 1992/93 and 2019/21 [24, 25]. Nevertheless, the
COVID-19 pandemic may affect the fertility trends in India and its states through the changes
in the fertility behaviours resulting from the disruptions in the trends in lifestyles, livelihoods,

and health care services in the country. Therefore, it might be worth focusing on how exactly the COVID-19 pandemic impacted fertility behaviours in India.

Although fertility behaviours can be influenced by the COVID-19 pandemic, only a few studies were found to investigate that influence using empirical data. Moreover, those studies predominantly focused on high-income countries. A study showed that fertility intentions in the United Kingdom, Germany, France, Italy, and Spain dropped during the COVID-19 pandemic, with intergroup variations in that drop [27]. In Germany, the falls in fertility intentions were a little more concentrated in the regions with the highest infection rates, whereas that fall in Italy was concentrated among those who did not have tertiary education and were younger than 30. In the United Kingdom, cancellations of fertility intentions were most frequent among those who believed that the pandemic might dramatically reduce their income. A study in New York City found that mothers with young children were frequently cancelling their pregnancy intentions during the first few months of the COVID-19 pandemic [27]. Another study in the USA showed that 34% of women postponed their pregnancy plans because of the pandemic [28].

A study in an upper middle-income country (Moldova) found that the individuals interviewed during the post-lockdown period were significantly less likely to try for conception and more likely to have sex [5]. In this study, the pandemic did not significantly influence modern contraceptive use. A study in Shanghai city cited that 33.8% of the participants cancelled their pregnancy plans because of the pandemic [29]. In Shanghai, mistrust of the government and health service system in managing the pandemic and concerns about the consequences of the pandemic for female and fetal health were associated with the cancellation of pregnancy plans.

A study based on the data from social media in Bangladesh–a lower middle-income country in South Asia–revealed that 20% of the respondents cancelled their COVID-19 pandemic period pregnancy plan because of the existing or potential health or socioeconomic crises related to the pandemic [30].

The studies that investigated the associations between the COVID-19 pandemic and fertility behaviours primarily focused in Europe and the United States of America, and the other parts are underrepresented in research. These studies also revealed that the participants negatively revised their pandemic period pregnancy plan. Various consequences of the pandemic in distinct socioeconomic and cultural settings may also influence fertility positively [31]. Nevertheless, there is not enough empirical research to reveal the differential effects of the pandemic on fertility behaviours. These facts lead us to the question: how the COVID-19 pandemic and associated lockdown reshaped fertility behaviours in a lower-middle income country? Our study intends to answer this question by investigating the changes in fertility behaviours after the COVID-10 lockdown in India, a country characterized by a diverse economy and culture. This study offers the opportunity to compare data from various states in India and may therefore provide important insight into the impact of the COVID-19 pandemic on fertility behaviours in different pandemic, socioeconomic, and demographic settings.

## Profiles of India and states

Out of the 36 states and union territories (UTs) of India, our study included 13 states/UTs that were interviewed during both pre- and post-lockdown periods. Out of these 13 states/UTS, Punjab, Uttarakhand, Haryana, Delhi (the National Capital Territory), and Rajasthan were from northern India, Uttar Pradesh, Madhya Pradesh, and Chhattisgarh were from central India, Jharkhand, and Odisha were from eastern India, Arunachal Pradesh was from Northeast India, and Tamil Nadu and Puducherry were from southern India. The Socio-economic, demographic, and COVID-19 pandemic profiles of India and its states are presented in the following subsections.

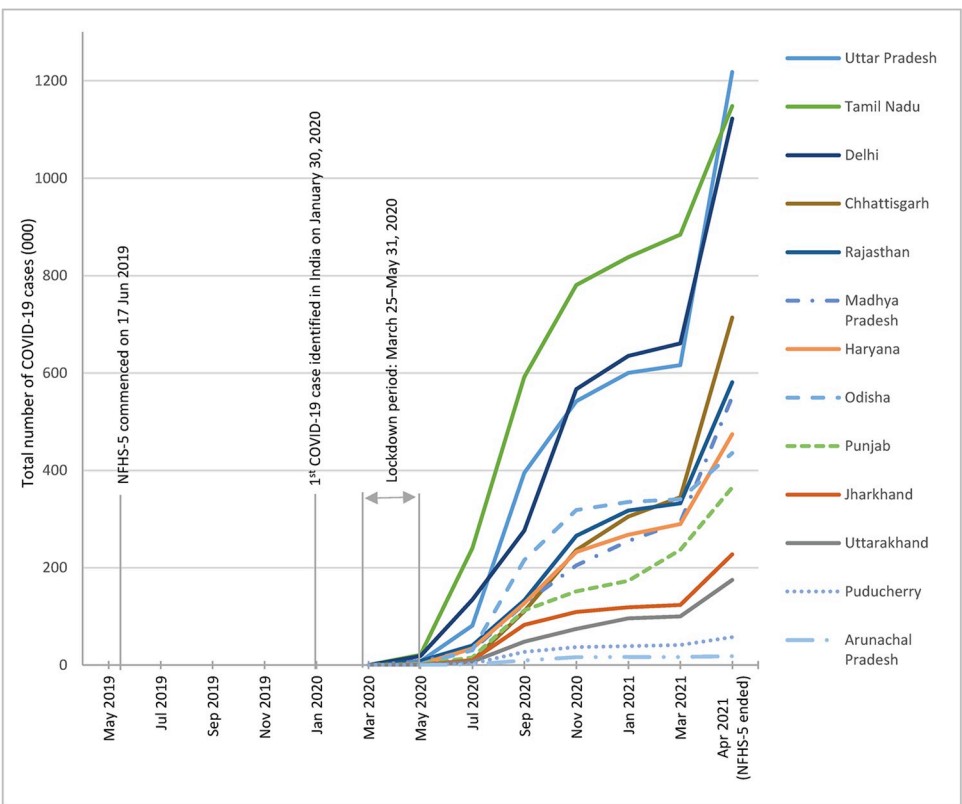

**Fig 1. Trends in total COVID-19 cases in the selected Indian states during the fieldwork of NFHS-5 2019/21.**
Source: PRS Legislative Research. Number of Cases. 2022 [32].

## COVID-19 pandemic profiles

After the first identification of a novel coronavirus in December 2019 in Wuhan, China, the virus spread all over the world very quickly. The trends in total COVID-19 cases by Indian states have been presented in Fig 1. The first Coronavirus positive case in India was reported approximately seven and a half months after the commencement of the fieldwork (on June 17, 2019) of the National Family Health Survey-5 (NFHS-5) 2019/21 that furnished the data for the current study [2, 24]. In India, the first Coronavirus positive case was reported in Kerala state on January 30, 2020 [2]. The virus had left its footprint in almost all of the Indian states and UTs (33 out of 36 states and UTs) by mid-April 2020 [32]. Three Indian states/UTs (Dadra and Nagar Haveli and Daman and Diu, Lakshadweep, and Sikkim) saw a coronavirus positive case later [32]. Before the lockdown was imposed on March 25, 2020, a total of 492 COVID-19 cases were identified in India, and that number rose to 18.8 million by the end of the fieldwork of the NFHS-5 on April 30, 2021. Among the selected states, Uttar Pradesh was the hardest hit by the pandemic (had a total of 1.2 million COVID-19 cases by the end of the survey), and Arunachal Pradesh had the lowest number of cases (18,256 cases) by the end of the survey [30].

## Socioeconomic profiles

The predicted differential effects of the COVID-19 pandemic on fertility in the countries with different levels of socioeconomic development make it important to discuss the socioeconomic development status of India and its states/UTs [31]. Currently, India is a lower middle income

country [20]. According to the baseline report on the National Multidimensional Poverty Index-2021 (NMPI-2021), approximately one out of every four Indians is multidimensionally poor [33]. The NMPI-2021 report used 12 indicators from three dimensions (health, education, and standard of living) for calculating the multidimensional poverty rate (MPR) and multidimensional poverty index (MPI). The NMPI-2021 report classified the states and UTs into five groups using the MPI values [33]. Using that classification, we further classified the socioeconomic status of the states and UTs into three groups: i) rich socioeconomic status having an MPI value of 0.105 or lower; ii) moderate socioeconomic status with an MPI value between 0.106 and 0.158; and iii) poor socioeconomic status having an MPI value of 0.159 or higher. According to this classification among the 13 states selected for this study, the socioeconomic status of Punjab (MPR = 5.6%), Tamil Nadu (MPR = 4.9%), Delhi (MPR = 4.8%), Puducherry (MPR = 1.7%), Uttarakhand (MPR = 17.7%), and Haryana (MPR = 12.3%) was rich; the socioeconomic status of Chhattisgarh (MPR = 29.9%), Rajasthan (MPR = 29.5%), Odisha (MPR = 29.4%), and Arunachal Pradesh (MPR = 24.3%) was moderate; and the socioeconomic status of Jharkhand (MPR = 42.2%), Uttar Pradesh (MPR = 37.8%), and Madhya Pradesh (MPR = 36.7%) was poor.

## Demographic profiles

India is the second-most populous country in the world with an 1.4 billion people [21]. The selected states and UTs comprise over 50% of the total Indian population, with Uttar Pradesh having the highest population (237.9 million) and Puducherry having the lowest population (1.4 million) among those selected states and UTs [34]. The population growth rate in India has decreased substantially during the past few decades, primarily because of its fertility decline [35].

India and all of its states have experienced noticeable fertility declines over the last three decades. The TFR in India decreased to 2.0 in 2019/21 from 3.4 in 1992/93 [23, 24]. TFR fell below two in all the selected states/UTs except Rajasthan, Uttar Pradesh, and Jharkhand. The TFRs in the two selected southern states/UTs were lower compared to other states/UTs. TFR in India decreased during the recent inter-NFHS period (2015/16–2019/21). During this inter-survey period, the TFRs in all the selected states also declined, except in Punjab and Tamil Nadu. Among the selected states and UTs, Uttar Pradesh had the highest TFR in 2019/21, while Punjab had the lowest TFR during the same period. This decline in fertility resulted from changes in the fertility behaviours of the people. The mean ideal family size in India decreased to 2.1 in 2019/21 from 2.9 in 1992/93. During this period, all the Indian states experienced substantial declines in their ideal family sizes. Among the states selected for this study, the mean ideal family sizes increased in Arunachal Pradesh, Madhya Pradesh, and Tamil Nadu, while the mean ideal family sizes stalled in Haryana, Delhi, and Puducherry during the recent inter-NFHS period (2015/16–2019/21). The remaining selected states experienced decreases in the ideal family sizes during that period. Contraceptive use in India increased to 66.7% in 2019/21 from 40.6% in 1992/93. Contraceptive uses in all of the selected states also increased between 1992/93 and 2019/21 [23, 24, 36].

## Methodology

### Data

This study used secondary data from the NFHS-5 conducted in 2019/21. The NFHS-5 data are freely available on the demographic and health survey (DHS) website (https://dhsprogram.com/data/dataset/India_Standard-DHS_2020.cfm?flag=1) upon a request for accessing the data. The NFHS-5 is a cross-sectional, nationally representative survey of 636,699 households,

providing information for a wide range of monitoring and impact evaluation indicators of health, reproduction, nutrition, disability, and women's empowerment. This study used the data of ever-married women of reproductive age (15–49 years). A woman was interviewed only if she provided verbal consent to participate in the survey in response to a loud reading of an informed consent statement by the interviewer. The NFHS-5 was approved by the Institutional Review Board of ICF (project number: 190308.0.001.00). The NFHS-5 adopted a uniform stratified two-stage sample design considering urban-rural area, village population sizes, percentages of the scheduled caste and scheduled tribe populations, and literacy rates. Probability proportional to size (PPS) sampling was used to select the primary sampling units (PSUs) or clusters. Villages in rural areas and Census Enumeration Blocks in urban areas were considered clusters. The 2011 census was used as the sampling frame for selecting the clusters. Trained research investigators used computer-assisted personal interviewing (CAPI) to collect data. For ensuring data quality, NFHS strictly followed the protocol for maintaining gender-sensitive team composition with a mix of an adequate number of male and female investigators who were from the native states and proficient in local languages [37]. NFHS ensured data quality by monitoring field work at multi-layer including spot checks, backchecks, review of field check tables, continuous feedback, debriefing field workers, and continuous supportive supervision [38]. The NFHS-5 received technical support from the International Institute for Population Sciences (IIPS), Mumbai, and the United States Agency for International Development (USAID) via ICF [38]. Field staff from IIPS conducted spot checks and back checks of surveyed households in a minimum of 10% of PSUs that were randomly selected by the IIPS central office [38]. The published reports contains more detailed descriptions of the questionnaire, quality control measures, survey design, and survey management procedure [24, 37, 38].

The NFHS-5 interviewed reproductive age ever-married women during the period between June 17, 2019 and April 30, 2021, with a short pause due to the COVID-19 lockdown. The lockdown lasted from March 25, 2020, to May 31, 2020. No fieldwork was done during the lockdown period. From the 36 states and UTs (28 states and 8 UTs) in India, interviews of 724,115 ever-married reproductive-aged (15–49 years) women (cases) from 30,161 clusters were completed with a 97% response rate. Out of these total interviewed cases, our study initially selected 358,603 weighted cases only from those 13 states and UTs (11 states and two UTs) that were surveyed during both pre- and post-lockdown periods (see Fig 2).

Out of those 358,603 cases, respectively, 42.9% and 57.1% were interviewed during pre- and post-lockdown periods. The selected 13 states and UTs constituted about half of the samples that were interviewed in India by the NFHS-5. For analysing the desire for a birth within two years, current contraceptive use, and sexual activities in the last 30 days in the pool of the 13 selected states and UTs, respectively, 56,561, 250,810, and 53,212 weighted cases (weighted using national-level sample weights) were selected following same selection criteria as the NFHS-5 published report (presented in Fig 3). The cases selected for analysing current contraceptive use excluded those who were not in a union or living with a man (including those who were never in a union) at the time of the survey, and who were not sexually active at the time of the survey. In selecting cases for analysing current contraceptive use, a woman was considered sexually inactive if currently she was not in a union or not living with a man and also did not have sex in the last 30 days preceding the survey. The selected cases for analysing desire for a birth within two years, current contraceptive use, and having sex in the last 30 days covered, respectively, 14,966, 15,645, and 4,712 clusters.

Besides the data on the ever-married women, the NFHS-5 also collected the geographic locations of the centers of the clusters. The geographic locations of the clusters that contributed to the selected samples were also analysed to examine the overall patterns of the geographic locations of the collected data.

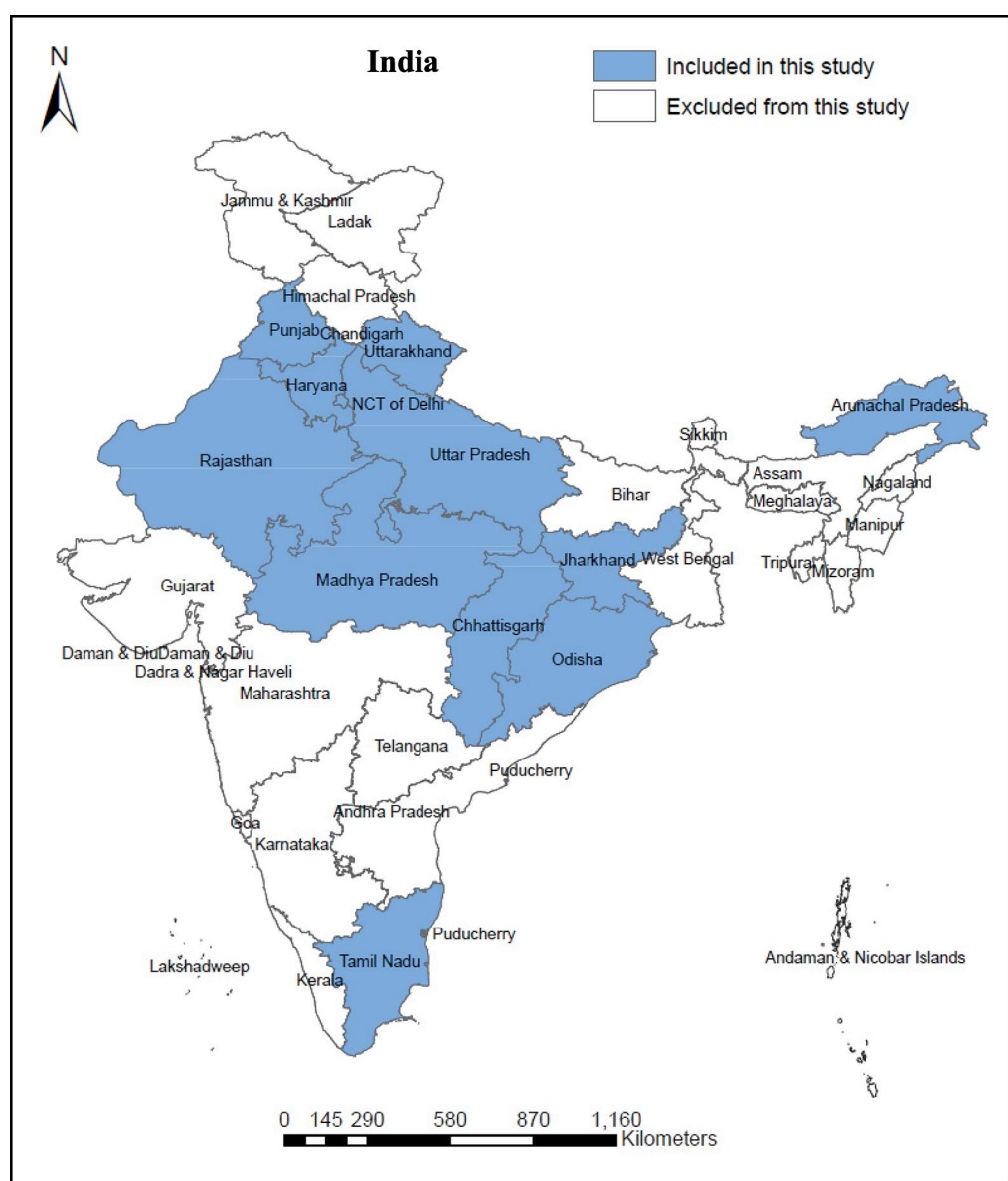

**Fig 2. States and UTs included in the study, India.** Source: Authors' production using the shapefile from Runfola et al. [39].

## Variables

The main objective of this study was to examine the differences between fertility related behaviours during the pre- and post-lockdown periods in India. For achieving this objective, we selected the following three response (dependent) variables based on the literature review and availability of data: i) desire for a birth within two years (before completion of two years), ii) current contraceptive use, and iii) having sex in the last 30 days. In selecting cases for each of the dependent variables, we followed the same criteria as the NFHS-5 published report. In analysing the desire for another birth, we excluded the women who did not want any more children and who gave non-numeric responses (for example, "don't know" and other non-numeric responses). In the analysis, those who desired a birth within two years were coded 1,

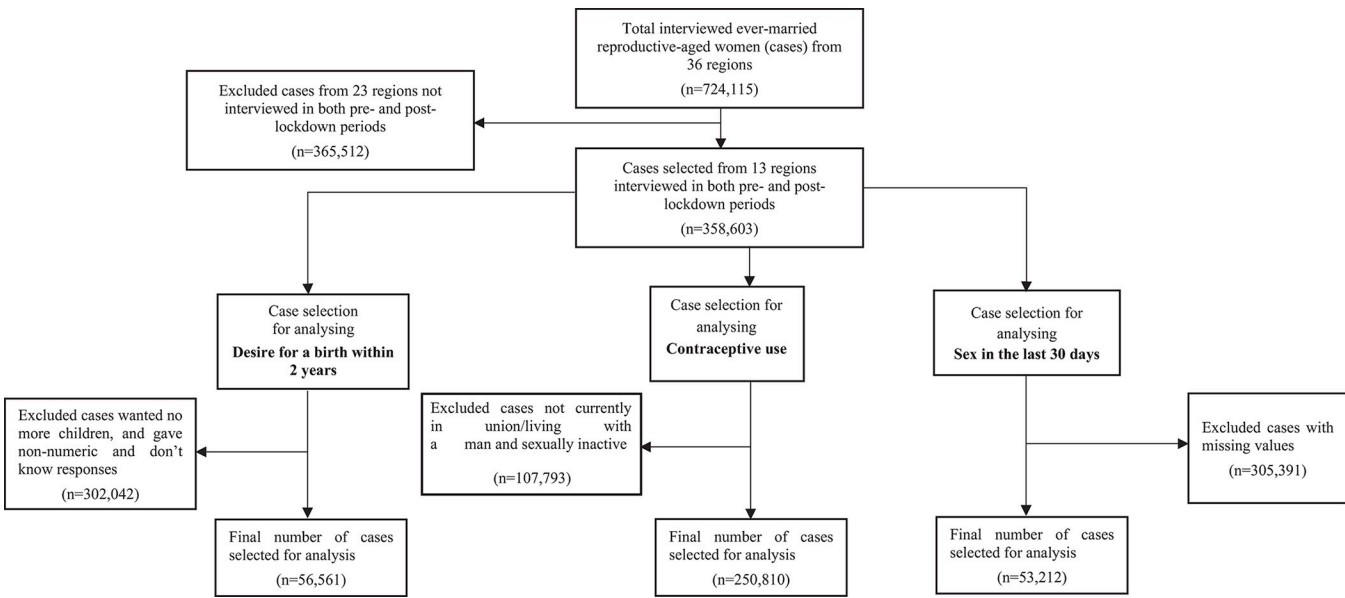

**Fig 3. Flowchart of case selection for analyses.** Note: All the numbers presented in the diagram were weighted using national level sample weights.

and those who desired a birth after two or more years were coded 0. Contraceptive use included both modern and traditional methods. For analysing contraceptive use, we included only the women who were currently in union, living with a man, or sexually active. The analysis of sex in the last 30 days was based on the women who reported the times since their last sexual intercourse.

The selection of the predictors (independent variables) for this study was guided by the literature review and availability of data. The predictors included in this study were whether a respondent was interviewed in the pre- or post-lockdown period, (respondent's) current age, number of children (alive), religion, caste, level of schooling, wealth status, residence, and media access.

The scheduled castes and tribes are the groups of people officially designated by the Constitution of India. The Government of India uses other backward class as a collective term to classify castes that are socially or educationally underprivileged [40]. Wealth status of a respondent was determined using the wealth index that was readily provided by the NFHS-5. The wealth index was created using scores based on the types and number of consumer goods owned and the housing characteristics. Principle component analysis was used to calculate the scores. Detailed methodology of calculating the wealth score is available elsewhere [41]. After ranking the respondents using their wealth scores, their distribution was divided into five equal groups, which were classified as poorest (in the lowest quintile), poorer (in the second quintile), middle (in the third quintile), richer (in the fourth quintile), and richest (in the highest quintile). A woman was considered to have media access if she accessed at least one of the following media at least once a week: television, radio, a newspaper, and a magazine.

## Statistical analysis

Both descriptive and multivariate analyses were performed to analyse the data. All these analyses were performed using the technique developed for complex sample design, which uses sampling strata, clusters, and sample weights. This technique is very important for estimating accurate confidence intervals and significance levels [42]. State-level sample weights were

applied to analyse the data from individual states and UTs, and the pooled data were analysed by applying national-level sample weights. In descriptive analysis, percentage distributions and 95% confidence intervals were used to examine the differences between fertility behaviours in the pre- and post-lockdown periods.

For multivariate analysis, we used a two-level mixed effects logit model to address the hierarchical composition of the NFHS-5 data. This two-level model was used because the individuals (level 1) were nested within clusters (level 2). The mixed-effects model can be preferred because it can incorporate the variability in the estimates of the effects of the predictors over the clusters, even if they are small [43]. Estimates of F-statistic and intraclass correlation coefficient (ICC) were used to examine the goodness of the fits of the models. The ICC accounts for the extent of similarity of the outcomes in each cluster that may result from a random effect compared to the outcomes in other clusters. An ICC value of 0.01, 0.10, and 0.25 might be considered a small, medium, or large effect, respectively [44, 45]. A mixed-effects logit regression reduces to a standard logit regression in the absence of a random effect [46].

A dichotomous predictor–interview period (whether interviewed in the pre- or post-lockdown period)–was included in the model to examine the effects of the pandemic lockdown on fertility-related behaviours when controlled for the effects of other relevant predictors. A dichotomous variable was chosen instead of a categorical month variable because a very small number of women were interviewed in some months. The effects of the lockdown on each fertility variable were examined for all the selected states, UTs, and pooled data. Besides these, the geographic pattern of sample collection was examined by simply plotting the GPS locations of the clusters to which the selected cases belonged.

The descriptive analysis was performed using IBM SPSS Statistics for Windows, version 23.0 (IBM Corp., Armonk, N.Y., USA), and the multivariate analysis was performed using Stata Statistical Software for Windows, release 17.0 (StataCorp LLC, College Station, Texas, USA). The geographic locations of the clusters were plotted using ArcGIS Desktop for Windows, version 10.7.1 (ESRI, Redlands, C.A., USA).

## Profiles of samples and interviews

Fig 4 shows the geographic locations of the clusters that contributed cases to the analyses in this study. All the selected states were interviewed in both pre- and post-lockdown periods, and the geographic locations of the overall clusters by interview period (pre- and post-lockdown period) did not show any systematic pattern regarding geographic direction (Fig 4).

Comparatively more women were interviewed in the post-lockdown period. Among the cases selected for analysing the desire for a birth within two years, 43.1% were interviewed in the pre-lockdown period, and the remaining were interviewed in the post-lockdown period. In analysing contraceptive use, 43.3% of the selected cases were interviewed in the pre-lockdown period, and the percentage for those selected for analysing having sex in the last 30 days was 41.5%. Selected weighted cases by state and UT can be found in the supplementary S1 Table.

Among the women interviewed in the pre- and post-lockdown periods, more than 80% were Hindu. This high percentage of Hindu respondents was because of the fact that around 79% of people in India are Hindu [47]. In both pre- and post-lockdown periods, the survey interviewed around 70% of women from rural areas. The distributions of the selected cases among the subcategories of the characteristics in the pre- and post-lockdown periods showed similar trends. Detailed data on the cases with different background characteristics by interview period can be found in the supplementary S2 Table.

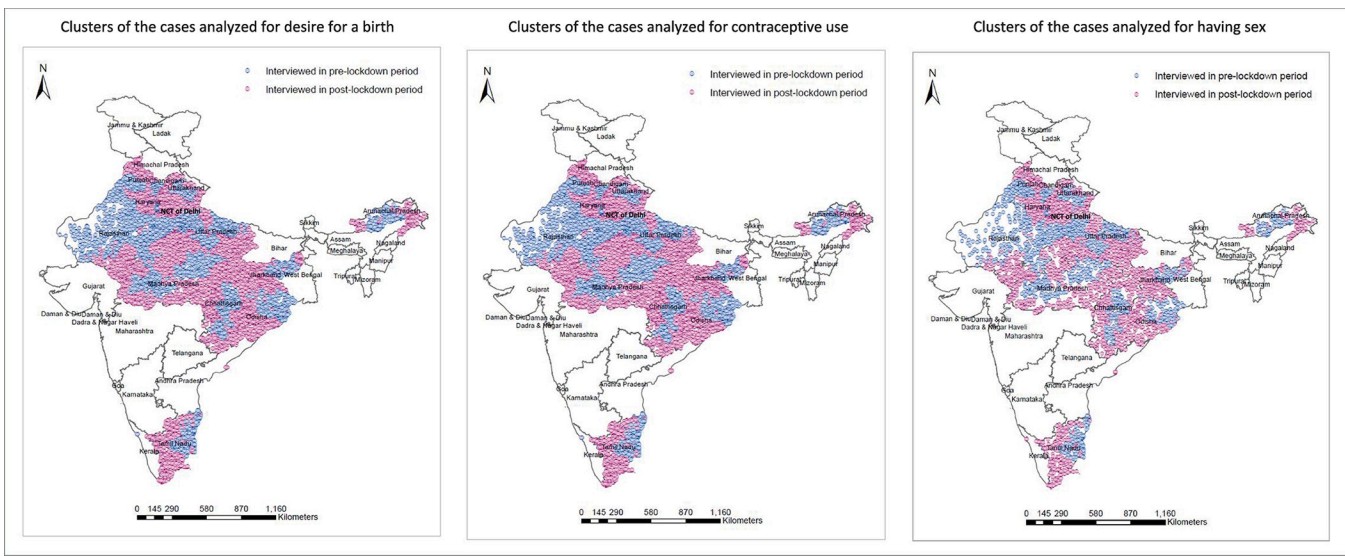

**Fig 4. Clusters of the included cases by interview period, India, 2019/21.** Source: Authors' production using the shapefile from 2019/21 NFHS-5 and Runfola et al. [39].

## Results

### Results of the descriptive analysis

Table 1 shows that the percentage of the selected women desiring a birth within two years in the pooled data substantially increased to 52.4% after the lockdown from 48.6% recorded prior to the lockdown. Among the 13 selected regions, nine regions experienced an increase in the percentage of women desiring a birth within two years, while that percentage declined in the

**Table 1. Fertility-related behaviours of the women by interview period in the selected Indian states, 2019/21.**

| State/union territory | Desire for a birth within 2 years % | | Contraceptive use % | | Had sex in last 30 days % | |
|---|---|---|---|---|---|---|
| | [95% CI] | | [95% CI] | | [95% CI] | |
| | Pre-lockdown | Post-lockdown | Pre-lockdown | Post-lockdown | Pre-lockdown | Post-lockdown |
| Pooled data | 48.6 [47.8–49.4] | 52.4 [51.7–53.2] | 68.8 [68.3–69.2] | 67.0 [66.6–67.3] | 37.3 [36.5–38.2] | 32.7 [31.9–33.5] |
| Punjab | 64.9 [61.3–68.3] | 63.8 [60.6–66.8] | 63.0 [60.7–65.3] | 68.5 [66.9–70.0] | 43.3 [40.6–45.9] | 33.8 [30.9–36.8] |
| Uttarakhand | 54.0 [48.4–59.5] | 53.9 [49.4–58.4] | 70.3 [68.2–72.3] | 71.1 [69.0–73.2] | 29.6 [25.2–34.4] | 33.3 [29.9–36.8] |
| Haryana | 53.4 [49.7–57.1] | 52.1 [49.5–54.8] | 75.1 [73.5–76.7] | 72.4 [71.2–73.6] | 41.5 [37.1–45.9] | 46.2 [43.7–48.7] |
| Delhi | 55.8 [51.6–59.9] | 52.5 [47.9–57.0] | 73.0 [71.0–74.9] | 81.7 [79.9–83.4] | 33.7 [29.3–38.4] | 32.1 [28.9–35.5] |
| Rajasthan | 45.1 [43.4–46.9] | 50.5 [47.5–53.4] | 72.4 [71.4–73.4] | 71.9 [70.4–73.4] | 44.5 [42.7–46.3] | 41.6 [38.1–45.1] |
| Uttar Pradesh | 46.0 [44.5–47.6] | 47.6 [46.3–48.9] | 63.3 [62.3–64.2] | 61.9 [61.2–62.6] | 35.3 [33.6–36.9] | 33.2 [31.9–34.6] |
| Arunachal Pradesh | 47.5 [43.5–51.6] | 51.1 [48.0–54.2] | 52.3 [49.5–55.1] | 65.0 [62.9–67.0] | 33.2 [28.9–37.8] | 37.5 [34.3–40.8] |
| Jharkhand | 47.1 [44.1–50.2] | 51.2 [48.8–53.6] | 67.1 [65.4–68.7] | 58.0 [56.7–59.3] | 40.3 [36.6–44.1] | 36.4 [33.9–38.9] |
| Odisha | 43.9 [40.6–47.2] | 45.1 [43.1–47.2] | 75.7 [74.3–77.1] | 72.8 [71.6–74.1] | 33.7 [30.7–36.8] | 32.1 [30.1–34.2] |
| Chhattisgarh | 52.5 [49.5–55.5] | 54.9 [52.3–57.5] | 68.6 [66.6–70.4] | 67.2 [65.8–68.6] | 35.3 [31.5–39.4] | 35.2 [31.8–38.8] |
| Madhya Pradesh | 47.7 [45.4–50.0] | 50.0 [48.2–51.7] | 70.6 [69.3–71.9] | 72.5 [71.7–73.4] | 46.3 [43.9–48.7] | 39.9 [38.1–41.7] |
| Tamil Nadu | 61.8 [58.4–65.0] | 68.5 [66.1–70.8] | 68.3 [66.6–70.0] | 68.7 [67.5–69.8] | 17.7 [15.1–20.6] | 15.1 [13.3–17.0] |
| Puducherry | 63.3 [53.7–72.0] | 81.0 [59.4–92.6] | 65.4 [60.9–69.6] | 69.7 [63.6–75.2] | 9.6 [5.5–16.2] | 15.6 [3.8–46.2] |

Source: Authors' calculation using NFHS-5, 2019/21.

Note: Confidence intervals were calculated using a logit transform.

remaining regions. Among the selected regions, the percentages of women desiring a birth within two years in Rajasthan and Tamil Nadu increased substantially after the lockdown.

The percentage of the selected women using any contraception in the pooled data substantially declined from 68.8% in the pre-lockdown period to 67% in the post-lockdown period (Table 1). The contraceptive use rate in six of the selected regions declined after the lockdown, while that rate increased in the remaining seven regions after the lockdown. The contraceptive use rate in Jharkhand and Odisha declined substantially after the lockdown, whereas the rate in Punjab, Delhi, and Arunachal Pradesh increased substantially after the lockdown.

As it is seen in Table 1, the rate of women having sex in the last 30 days among those interviewed after the lockdown (32.7%) in the pooled data was noticeably lower than that among those interviewed before the lockdown (37.3%). This rate in the nine selected regions declined after the lockdown, while that in the remaining four states increased after the lockdown. Among the selected states, the rates of having sex in Punjab and Madhya Pradesh substantially declined after the lockdown.

## Results of the mixed effects logit model

Mixed-effects logit models were fitted to estimate the net effects of the COVID-19 lockdown on fertility-related factors by controlling for relevant variables. It is seen from Table 2 that the odds of desiring another birth within two years among the pooled women interviewed in the post-lockdown period (OR = 1.20, p<0.001) was 20% higher than that among those interviewed in the pre-lockdown period. The women interviewed in the post-lockdown period in Rajasthan (OR = 1.18, p<0.05), Arunachal Pradesh (OR = 1.41, p<0.05), Chhattisgarh (OR = 1.26, p<0.05), and Tamil Nadu (OR = 1.60, p<0.001) were also significantly more likely to desire another birth within two years than those who were interviewed in the pre-lockdown period in those regions (Table 2).

Table 3 shows that in the pooled data, the odds of using contraception in the post-lockdown period (OR = 0.92, p<0.001) was 8.4% lower than that in the pre-lockdown period. The odds of using contraception in the post-lockdown period in Jharkhand (OR = 0.70, p<0.001), Odisha (OR = 0.85, p<0.01), and Uttar Pradesh (OR = 0.90, p<0.001) were significantly lower than that in the pre-lockdown period. While women interviewed in the post-lockdown period in Arunachal Pradesh (OR = 1.79, p<0.001), Delhi (OR = 1.92, p<0.001), Punjab (OR = 1.50, p<0.001), and Tamil Nadu (OR = 1.14, p<0.05) were significantly more likely to use contraception than those interviewed in the pre-lockdown period (Table 3).

Among the pooled women, the odds of having sex (in the last 30 days) in the post-lockdown period (OR = 0.78, p<0.001) was 22% lower than that in the pre-lockdown period (Table 4). Women in Punjab (OR = 0.64, p<0.001), Uttar Pradesh (OR = 0.87, p<0.05), Madhya Pradesh (OR = 0.70, p<0.001), and Tamil Nadu (OR = 0.75, p<0.05) were significantly less likely to have sex in the post-lockdown period than in the pre-lockdown period. In contrast, women in Haryana (OR = 1.43, p<0.01) were significantly more likely to have sex in the post-lockdown period than in the pre-lockdown period.

The F-statistics of all the fitted models that have been presented in Tables 2–4 are statistically significant, indicating the overall significance of all of the fitted models. Table 2 shows the cluster memberships in pooled data (ICC = 0.117), Uttarakhand (ICC = 0.0.135), Arunachal Pradesh (ICC = 0.225), Jharkhand (ICC = 0.137), Chhattisgarh (ICC = 0.128), and Madhya Pradesh (ICC = 0.072) showed a medium effect on desire for another birth, and that effects in the remaining states/UTs were small.

Table 3 shows that the effects of the cluster memberships on using contraception in pooled data (ICC = 0.138), Punjab (ICC = 0.153), Rajasthan (ICC = 0.167), Uttar Pradesh

**Table 2. Adjusted odds ratios (ORs) of desiring a birth within 2 years among the women interviewed in pre- and post-lockdown periods in the selected Indian states, 2019/21.**

| Characteristics | Pooled data | Punjab | Uttarakhand | Haryana | Delhi | Rajasthan | Uttar Pradesh | Arunachal Pradesh | Jharkhand | Odisha | Chhattisgarh | Madhya Pradesh | Tamil Nadu | Puducherry |
|---|---|---|---|---|---|---|---|---|---|---|---|---|---|---|
| | N = 56,561 | n = 12,650 | n = 1,783 | n = 2,925 | n = 1,525 | n = 7,579 | n = 15,088 | n = 3,041 | n = 4,813 | n = 4,605 | n = 4,573 | n = 7,241 | n = 3,838 | n = 463 |
| **Interviewed in** | | | | | | | | | | | | | | |
| Pre-lockdown (R) | 1.00 | 1.00 | 1.00 | 1.00 | 1.00 | 1.00 | 1.00 | 1.00 | 1.00 | 1.00 | 1.00 | 1.00 | 1.00 | 1.00 |
| Post-lockdown | 1.20*** | 0.97 | 1.00 | 0.90 | 0.84 | 1.18* | 1.08 | 1.41* | 1.14 | 1.06 | 1.26* | 1.11 | 1.60*** | 4.17 |
| **Current age** | | | | | | | | | | | | | | |
| 15–24 (R) | 1.00 | 1.00 | 1.00 | 1.00 | 1.00 | 1.00 | 1.00 | 1.00 | 1.00 | 1.00 | 1.00 | 1.00 | 1.00 | 1.00 |
| 25–34 | 2.58*** | 2.87*** | 3.81*** | 2.61*** | 2.88*** | 2.73*** | 2.35*** | 2.51*** | 2.61*** | 3.60*** | 2.03*** | 2.53*** | 2.28*** | 2.20* |
| 35–49 | 8.18*** | 6.75*** | 12.76*** | 14.43*** | 19.49*** | 7.75*** | 6.86*** | 5.12*** | 6.71*** | 13.19*** | 5.85*** | 6.17*** | 8.76*** | 5.46* |
| **Number of children** | | | | | | | | | | | | | | |
| 0 (R) | 1.00 | 1.00 | 1.00 | 1.00 | 1.00 | 1.00 | 1.00 | 1.00 | 1.00 | 1.00 | 1.00 | 1.00 | 1.00 | 1.00 |
| 1–2 | 0.22*** | 0.18*** | 0.17*** | 0.19*** | 0.18*** | 0.30*** | 0.20*** | 0.43*** | 0.20*** | 0.14*** | 0.21*** | 0.29*** | 0.23*** | 0.21*** |
| >2 | 0.20*** | 0.40* | 0.10*** | 0.13*** | 0.14*** | 0.31*** | 0.20*** | 0.40*** | 0.16*** | 0.07*** | 0.20*** | 0.27*** | 0.11*** | 3.25 |
| **Religion** | | | | | | | | | | | | | | |
| Hindu (R) | 1.00 | 1.00 | 1.00 | 1.00 | 1.00 | 1.00 | 1.00 | 1.00 | 1.00 | 1.00 | 1.00 | 1.00 | 1.00 | 1.00 |
| Muslim | 0.89** | 1.24 | 1.28 | 1.20 | 0.80 | 0.93 | 1.02 | 0.80 | 0.89 | 0.89 | 0.80 | 0.76* | 0.78 | 0.21** |
| Christian | 1.26* | 1.04 | 1.00 | 0.96 | 0.24 | 2.49 | 0.91 | 0.99 | 1.03 | 0.84 | 1.08 | 1.66 | 0.97 | 6.02 |
| Other | 1.34*** | 1.01 | 2.12 | 1.29 | 1.65 | 1.01 | 1.05 | 1.04 | 1.16 | 1.00 | 1.18 | 1.20 | - | - |
| **Caste** | | | | | | | | | | | | | | |
| Scheduled caste/tribe (R) | 1.00 | 1.00 | 1.00 | 1.00 | 1.00 | 1.00 | 1.00 | 1.00 | 1.00 | 1.00 | 1.00 | 1.00 | 1.00 | 1.00 |
| Other backward classes | 1.02 | 1.39 | 0.71 | 1.09 | 0.84 | 0.93 | 1.03 | 0.47** | 1.00 | 1.13 | 1.05 | 1.05 | 1.04 | 1.90 |
| Other | 0.90** | 1.12 | 0.76 | 1.21 | 0.92 | 1.00 | 1.02 | 0.41*** | 1.11 | 0.97 | 1.81 | 1.06 | 0.53 | 0.83 |
| Don't know/ missing | 1.39*** | 5.89*** | 0.65 | 0.52 | 1.17 | 1.33 | 1.68 | 0.71 | 1.48 | 2.45* | 0.86 | 1.45 | 1.10 | 1.00 |
| **Level of schooling** | | | | | | | | | | | | | | |
| No schooling (R) | 1.00 | 1.00 | 1.00 | 1.00 | 1.00 | 1.00 | 1.00 | 1.00 | 1.00 | 1.00 | 1.00 | 1.00 | 1.00 | 1.00 |
| <7 years complete | 0.93 | 0.73 | 0.98 | 0.70 | 0.84 | 0.98 | 0.87 | 0.91 | 0.81 | 0.70* | 0.99 | 1.05 | 1.67 | 0.26 |
| 7–9 years complete | 0.76*** | 0.55 | 0.77 | 0.64 | 0.66 | 0.74** | 0.72*** | 0.63* | 0.75** | 0.74* | 0.67** | 0.69*** | 1.33 | 0.72 |
| 10–11 years complete | 0.71*** | 0.53* | 0.66 | 0.48** | 0.88 | 0.71** | 0.68*** | 0.59** | 0.83 | 0.53*** | 0.63** | 0.66** | 0.77 | 0.96 |
| 12+ years complete | 0.55*** | 0.31*** | 0.42* | 0.41*** | 0.44** | 0.58*** | 0.54*** | 0.68* | 0.57*** | 0.47* | 0.44*** | 0.47*** | 0.54 | 0.63 |
| **Wealth status** | | | | | | | | | | | | | | |
| Poorest (R) | 1.00 | 1.00 | 1.00 | 1.00 | 1.00 | 1.00 | 1.00 | 1.00 | 1.00 | 1.00 | 1.00 | 1.00 | 1.00 | 1.00 |
| Poorer | 1.00 | 1.27 | 1.16 | 1.13 | 1.071.26 | 0.96 | 1.01 | 0.75 | 0.92 | 1.01 | 1.06 | 1.01 | 0.81 | 0.63 |

(Continued)

**Table 2.** (Continued)

| Characteristics | Pooled data | Punjab | Uttarakhand | Haryana | Delhi | Rajasthan | Uttar Pradesh | Arunachal Pradesh | Jharkhand | Odisha | Chhattisgarh | Madhya Pradesh | Tamil Nadu | Puducherry |
|---|---|---|---|---|---|---|---|---|---|---|---|---|---|---|
| | N = 56,561 | n = 12,650 | n = 1,783 | n = 2,925 | n = 1,525 | n = 7,579 | n = 15,088 | n = 3,041 | n = 4,813 | n = 4,605 | n = 4,573 | n = 7,241 | n = 3,838 | n = 463 |
| Middle | 1.05 | 1.55 | 1.07 | 1.44 | 1.17 | 0.91 | 0.99 | 1.16 | 0.88 | 0.88 | 0.94 | 0.93 | 1.03 | 1.37 |
| Richer | 1.04 | 1.07 | 1.79 | 1.12 | 1.00 | 0.88 | 1.03 | 0.89 | 0.70 | 0.96 | 1.11 | 0.87 | 0.90 | 0.53 |
| Richest | 1.06 | 1.42 | 1.18 | 1.12 | | 0.98 | 1.06 | 0.90 | 0.64 | 0.79 | 1.41 | 0.90 | 1.18 | 0.93 |
| Residence | | | | | | | | | | | | | | |
| Urban ® | 1.00 | 1.00 | 1.00 | 1.00 | 1.00 | 1.00 | 1.00 | 1.00 | 1.00 | 1.00 | 1.00 | 1.00 | 1.00 | 1.00 |
| Rural | 0.91** | 1.04 | 1.52 | 1.40** | 0.63 | 0.82* | 1.02 | 1.03 | 0.91 | 0.97 | 1.04 | 1.05 | 1.05 | 1.82 |
| Had media access | | | | | | | | | | | | | | |
| Yes ® | 1.00 | 1.00 | 1.00 | 1.00 | 1.00 | 1.00 | 1.00 | 1.00 | 1.00 | 1.00 | 1.00 | 1.00 | 1.00 | 1.00 |
| No | 0.90*** | 0.93 | 1.19 | 0.95 | 1.09 | 1.15* | 0.91 | 1.05 | 0.93 | 0.97 | 1.13 | 1.05 | 0.76* | 1.22 |
| Constant | 2.21*** | 3.87 | 2.16 | 2.51 | 3.07*** | 1.71** | 2.02*** | 1.31 | 2.96*** | 2.09*** | 2.59*** | 1.78*** | 3.49** | 2.50 |
| Model fit statistics | | | | | | | | | | | | | | |
| F-statistic | 195.3*** | 11.1*** | 8.4*** | 12.6*** | 10.3*** | 23.1*** | 65.2*** | 6.6*** | 19.8*** | 24.6*** | 16.9*** | 20.7*** | 17.5*** | 2.19** |
| ICC | 0.117 | 0.085 | 0.135 | 0.088 | 0.069 | 0.091 | 0.092 | 0.225 | 0.137 | 0.080 | 0.128 | 0.072 | 0.100 | 0.110 |

Source: Authors' calculation using NFHS-5, 2019/21.

Note: ICC = Intraclass correlation coefficient. The number of cases in pooled data (N) and in states' data (n) are weighted with, respectively, national level and state level weights. Supplementary S3 Table represents 95% confidence intervals and exact P-values of the estimates.

\* p < 0.05

\*\* p < 0.01

\*\*\* p < 0.001.

Table 3. Adjusted odds ratios (ORs) of using contraception among the women interviewed in pre- and post-lockdown periods in the selected Indian states, 2019/21.

| Characteristics | Pooled data | Punjab | Uttarakhand | Haryana | Delhi | Rajasthan | Uttar Pradesh | Arunachal Pradesh | Jharkhand | Odisha | Chhattisgarh | Madhya Pradesh | Tamil Nadu | Puducherry |
|---|---|---|---|---|---|---|---|---|---|---|---|---|---|---|
| | N = 250,810 | n = 15,349 | n = 9,151 | n = 15,745 | n = 7,437 | n = 30,785 | n = 62,700 | n = 13,658 | n = 19,483 | n = 20,183 | n = 18,927 | n = 35,003 | n = 18,475 | n = 2,436 |
| **Interviewed in** | | | | | | | | | | | | | | |
| Pre-lockdown ⓡ | 1.00 | 1.00 | 1.00 | 1.00 | 1.00 | 1.00 | 1.00 | 1.00 | 1.00 | 1.00 | 1.00 | 1.00 | 1.00 | 1.00 |
| Post-lockdown | 0.92*** | 1.50*** | 1.10 | 1.00 | 1.92*** | 1.00 | 0.90*** | 1.79*** | 0.70*** | 0.85* | 0.88 | 1.09 | 1.14* | 1.56 |
| **Current age** | | | | | | | | | | | | | | |
| 15–24 ⓡ | 1.00 | 1.00 | 1.00 | 1.00 | 1.00 | 1.00 | 1.00 | 1.00 | 1.00 | 1.00 | 1.00 | 1.00 | 1.00 | 1.00 |
| 25–34 | 1.85*** | 1.72*** | 1.89*** | 2.48*** | 1.65*** | 1.90*** | 1.56*** | 1.26* | 2.05*** | 1.75*** | 2.03*** | 2.62*** | 1.98*** | 1.87 |
| 35–49 | 2.33*** | 1.82*** | 2.39*** | 3.27*** | 1.51*** | 3.67*** | 1.47*** | 1.34*** | 2.51*** | 1.51*** | 3.70*** | 4.35*** | 3.40*** | 4.04*** |
| **Number of children** | | | | | | | | | | | | | | |
| 0 ⓡ | 1.00 | 1.00 | 1.00 | 1.00 | 1.00 | 1.00 | 1.00 | 1.00 | 1.00 | 1.00 | 1.00 | 1.00 | 1.00 | 1.00 |
| 1–2 | 10.37*** | 15.22*** | 9.16*** | 16.54*** | 11.31*** | 7.32*** | 6.10*** | 3.14****** | 11.12*** | 13.64*** | 27.00*** | 12.89*** | 107.75*** | 42.26*** |
| >2 | 16.44*** | 22.24*** | 12.70*** | 27.03*** | 18.93*** | 12.82*** | 10.00*** | 4.55*** | 20.07*** | 19.30*** | 58.36*** | 20.58*** | 293.53*** | 146.02*** |
| **Religion** | | | | | | | | | | | | | | |
| Hindu ⓡ | 1.00 | 1.00 | 1.00 | 1.00 | 1.00 | 1.00 | 1.00 | 1.00 | 1.00 | 1.00 | 1.00 | 1.00 | 1.00 | 1.00 |
| Muslim | 0.59*** | 0.47** | 0.69* | 0.35*** | 0.63*** | 0.67*** | 0.65*** | 0.82 | 0.44*** | 0.89 | 0.82 | 0.65*** | 0.63*** | 0.38* |
| Christian | 0.88* | 0.91 | 1.00 | 0.34 | 4.73** | 0.83 | 0.34 | 0.99 | 0.86 | 0.90 | 0.74 | 1.24 | 1.03 | 0.57 |
| Other | 0.92* | 1.06 | 1.38 | 0.94 | 1.02 | 1.10 | 1.30 | 0.93 | 1.03 | 1.95 | 1.03 | 1.15 | 4.65 | 1.12 |
| **Caste** | | | | | | | | | | | | | | |
| Scheduled caste/tribe ⓡ | 1.00 | 1.00 | 1.00 | 1.00 | 1.00 | 1.00 | 1.00 | 1.00 | 1.00 | 1.00 | 1.00 | 1.00 | 1.00 | 1.00 |
| Other backward classes | 1.02 | 0.88 | 0.85 | 1.10 | 1.03 | 1.05 | 1.02 | 0.84 | 1.28*** | 0.90 | 1.31*** | 1.05 | 1.06 | 0.89 |
| Other | 1.10*** | 1.16* | 0.97 | 1.19** | 1.02 | 1.10 | 1.12** | 1.63*** | 1.19* | 0.90 | 1.40* | 0.92 | 0.87 | 0.91 |
| Don't know/ missing | 1.03 | 0.86 | 0.73 | 1.23 | 0.84 | 0.99 | 1.11 | 1.02 | 1.24 | 0.71 | 1.58** | 0.83 | 1.50 | 0.61 |
| **Level of schooling** | | | | | | | | | | | | | | |
| No schooling ⓡ | 1.00 | 1.00 | 1.00 | 1.00 | 1.00 | 1.00 | 1.00 | 1.00 | 1.00 | 1.00 | 1.00 | 1.00 | 1.00 | 1.00 |
| <7 years complete | 1.13*** | 0.95 | 1.48** | 1.28** | 1.10 | 1.07 | 1.03 | 1.51*** | 1.18* | 1.12 | 1.19* | 1.01 | 1.15 | 1.00 |
| 7–9 years complete | 1.06 | 0.96 | 1.36* | 1.05 | 1.14 | 0.90 | 1.06 | 1.41*** | 1.16** | 1.20* | 1.20* | 0.84*** | 1.15 | 0.76 |
| 10–11 years complete | 1.00 | 0.93 | 1.18 | 0.98 | 1.10 | 0.86* | 1.05 | 1.22* | 1.00 | 1.21* | 1.20 | 0.80** | 1.02 | 0.78 |
| 12+ years complete | 0.90*** | 0.90 | 1.30 | 0.96 | 1.27* | 0.97 | 1.05 | 1.12 | 0.99 | 1.15 | 0.95 | 0.74*** | 0.66*** | 0.40 |
| **Wealth status** | | | | | | | | | | | | | | |
| Poorest ⓡ | 1.00 | 1.00 | 1.00 | 1.00 | 1.00 | 1.00 | 1.00 | 1.00 | 1.00 | 1.00 | 1.00 | 1.00 | 1.00 | 1.00 |
| Poorer | 1.14*** | 1.43 | 1.14 | 1.26 | 1.07 | 1.06 | 1.11*** | 0.99 | 1.39*** | 1.00 | 1.22** | 1.10* | 0.92 | 0.45 |

(Continued)

**Table 3.** (Continued)

| Characteristics | Pooled data | Punjab | Uttarakhand | Haryana | Delhi | Rajasthan | Uttar Pradesh | Arunachal Pradesh | Jharkhand | Odisha | Chhattisgarh | Madhya Pradesh | Tamil Nadu | Puducherry |
|---|---|---|---|---|---|---|---|---|---|---|---|---|---|---|
| | N = 250,810 | n = 15,349 | n = 9,151 | n = 15,745 | n = 7,437 | n = 30,785 | n = 62,700 | n = 13,658 | n = 19,483 | n = 20,183 | n = 18,927 | n = 35,003 | n = 18,475 | n = 2,436 |
| Middle | 1.20*** | 1.37 | 0.87 | 1.60** | 1.07 | 0.98 | 1.17*** | 0.96 | 1.44*** | 1.03 | 1.50*** | 1.16** | 0.99 | 0.37 |
| Richer | 1.22*** | 1.10 | 0.78 | 1.66** | 1.20 | 1.06 | 1.21*** | 0.83 | 1.64*** | 1.09 | 1.60*** | 1.09 | 0.99 | 0.34 |
| Richest | 1.26*** | 1.06 | 0.81 | 1.72*** | 1.21 | 1.06 | 1.36*** | 1.05 | 1.39** | 1.09 | 1.71*** | 1.29** | 0.83 | 0.34* |
| Residence | | | | | | | | | | | | | | |
| Urban ⓡ | 1.00 | 1.00 | 1.00 | 1.00 | 1.00 | 1.00 | 1.00 | 1.00 | 1.00 | 1.00 | 1.00 | 1.00 | 1.00 | 1.00 |
| Rural | 0.90*** | 0.88* | 0.77** | 1.13 | 0.89 | 0.90 | 0.78*** | 1.07 | 0.97 | 0.86 | 1.09 | 1.15* | 0.97 | 1.02 |
| Had media access | | | | | | | | | | | | | | |
| Yes ⓡ | 1.00 | 1.00 | 1.00 | 1.00 | 1.00 | 1.00 | 1.00 | 1.00 | 1.00 | 1.00 | 1.00 | 1.00 | 1.00 | 1.00 |
| No | 0.81*** | 0.75*** | 0.81* | 0.78*** | 0.78** | 0.86*** | 0.86*** | 0.73*** | 0.98 | 0.89* | 0.77*** | 0.76*** | 1.00 | 0.77 |
| Constant | 0.14*** | 0.08*** | 0.20*** | 0.05*** | 0.16 | 0.23 | 0.24*** | 0.26*** | 0.07*** | 0.24*** | 0.03*** | 0.09*** | 0.01*** | 0.11* |
| Model fit statistics | | | | | | | | | | | | | | |
| F-statistic | 726.2*** | 44.8*** | 31.9*** | 69.7*** | 31.1*** | 118.1*** | 170.1*** | 21.5*** | 72.6*** | 72.5*** | 65.3*** | 139.4*** | 74.1*** | 10.8*** |
| ICC | 0.138 | 0.153 | 0.060 | 0.089 | 0.099 | 0.167 | 0.120 | 0.190 | 0.080 | 0.120 | 0.176 | 0.174 | 0.091 | 0.077 |

Source: Authors' calculation using NFHS-5, 2019/21.

Note: ICC = Intraclass correlation coefficient. The number of cases in pooled data (N) and in states' data (n) are weighted with, respectively, national level and state level weights. Supplementary S4 Table represents 95% confidence intervals and exact P-values of the estimates.

* p < 0.05

** p < 0.01

*** p < 0.001.

**Table 4. Adjusted odds -ratios (ORs) of having sex in the last 30 days among the women interviewed in pre- and post-lockdown periods in the selected Indian states, 2019/21.**

| Characteristics | Pooled data | Punjab | Uttarakhand | Haryana | Delhi | Rajasthan | Uttar Pradesh | Arunachal Pradesh | Jharkhand | Odisha | Chhattisgarh | Madhya Pradesh | Tamil Nadu | Puducherry |
|---|---|---|---|---|---|---|---|---|---|---|---|---|---|---|
| | N = 53,212 | n = 3,508 | n = 1,764 | n = 3,208 | n = 1,705 | n = 6,541 | n = 13,651 | n = 2,933 | n = 4,048 | n = 4,340 | n = 4,265 | n = 6,803 | n = 3,794 | n = 582 |
| **Interviewed in** | | | | | | | | | | | | | | |
| Pre-lockdown (R) | 1.00 | 1.00 | 1.00 | 1.00 | 1.00 | 1.00 | 1.00 | 1.00 | 1.00 | 1.00 | 1.00 | 1.00 | 1.00 | 1.00 |
| Post-lockdown | 0.78*** | 0.64*** | 1.08 | 1.43** | 0.93 | 0.88 | 0.87* | 1.27 | 0.83 | 0.93 | 0.89 | 0.70*** | 0.75* | 0.88 |
| **Current age** | | | | | | | | | | | | | | |
| 15–24 (R) | 1.00 | 1.00 | 1.00 | 1.00 | 1.00 | 1.00 | 1.00 | 1.00 | 1.00 | 1.00 | 1.00 | 1.00 | 1.00 | 1.00 |
| 25–34 | 1.91*** | 2.40* | 1.56 | 1.70** | 2.53*** | 2.02*** | 1.75*** | 1.68** | 1.52*** | 1.75*** | 2.76*** | 2.43*** | 3.11*** | 3.40 |
| 35–49 | 0.93 | 0.90 | 1.04 | 0.83 | 1.08 | 1.06 | 1.05 | 0.91 | 1.01 | 0.98 | 1.53* | 1.14 | 1.50 | 0.71 |
| **Number of children** | | | | | | | | | | | | | | |
| 0 (R) | 1.00 | 1.00 | 1.00 | 1.00 | 1.00 | 1.00 | 1.00 | 1.00 | 1.00 | 1.00 | 1.00 | 1.00 | 1.00 | 1.00 |
| 1–2 | 6.55*** | 7.81*** | 8.43*** | 10.04*** | 4.75*** | 6.53*** | 7.60*** | 6.86*** | 9.09*** | 6.25*** | 8.09*** | 7.12*** | 1.76* | 5.23 |
| >2 | 6.87*** | 8.15*** | 6.63*** | 9.36*** | 5.50*** | 6.11*** | 7.57*** | 6.51*** | 8.57*** | 6.06*** | 6.32*** | 5.64*** | 1.66 | 7.01 |
| **Religion** | | | | | | | | | | | | | | |
| Hindu (R) | 1.00 | 1.00 | 1.00 | 1.00 | 1.00 | 1.00 | 1.00 | 1.00 | 1.00 | 1.00 | 1.00 | 1.00 | 1.00 | 1.00 |
| Muslim | 1.12* | 0.82 | 1.91* | 1.54* | 1.25 | 1.18 | 1.04 | 0.41* | 0.99 | 1.19 | 0.68 | 1.08 | 1.35 | 1.83 |
| Christian | 0.80 | 1.52 | 1.00 | - | 0.33 | 0.74 | 1.00 | 1.01 | 1.48 | 1.19 | 0.43 | 0.26 | 1.18 | 1.38 |
| Other | 1.06 | 0.88 | 1.09 | 1.31 | 1.83* | 0.91 | 0.98 | 1.16 | 1.02 | 2.16 | 1.59 | 1.14 | 1.00 | 1.00 |
| **Caste** | | | | | | | | | | | | | | |
| Scheduled caste/tribe (R) | 1.00 | 1.00 | 1.00 | 1.00 | 1.00 | 1.00 | 1.00 | 1.00 | 1.00 | 1.00 | 1.00 | 1.00 | 1.00 | 1.00 |
| Other backward classes | 0.91* | 0.97 | 1.03 | 1.00 | 0.97 | 0.91 | 0.89* | 1.39 | 0.71* | 0.82 | 1.30* | 1.06 | 0.92 | 0.95 |
| Other | 1.00 | 1.01 | 0.82 | 0.85 | 1.15 | 0.85 | 0.86 | 1.02 | 0.80 | 0.71* | 1.13 | 0.86 | 0.50 | 11.34 |
| Don't know/ missing | 1.00 | 1.17 | 1.17 | 1.13 | 0.91 | 0.86 | 0.90 | 1.77 | 0.73 | 0.81 | 1.04 | 1.02 | 0.51 | 1.00 |
| **Level of schooling** | | | | | | | | | | | | | | |
| No schooling (R) | 1.00 | 1.00 | 1.00 | 1.00 | 1.00 | 1.00 | 1.00 | 1.00 | 1.00 | 1.00 | 1.00 | 1.00 | 1.00 | 1.00 |
| <7 years complete | 0.98 | 1.06 | 1.67 | 1.21 | 0.93 | 1.29* | 1.10 | 1.21 | 1.14 | 0.98 | 1.35 | 0.97 | 0.85 | 0.37 |
| 7–9 years complete | 0.93 | 1.12 | 1.09 | 1.22 | 0.90 | 1.13 | 0.94 | 1.17 | 1.28 | 0.88 | 1.28 | 0.91 | 1.22 | 0.48 |
| 10–11 years complete | 0.88* | 1.56 | 1.23 | 1.21 | 1.01 | 0.91 | 0.83 | 1.04 | 1.32 | 1.12 | 1.55* | 0.70* | 1.12 | 1.68 |
| 12+ years complete | 0.91* | 1.26 | 1.04 | 1.38 | 0.94 | 1.01 | 0.95 | 1.12 | 1.21 | 0.93 | 1.17 | 0.88 | 1.14 | 0.33 |
| **Wealth status** | | | | | | | | | | | | | | |

*(Continued)*

**Table 4.** (Continued)

| Characteristics | Pooled data | Punjab | Uttarakhand | Haryana | Delhi | Rajasthan | Uttar Pradesh | Arunachal Pradesh | Jharkhand | Odisha | Chhattisgarh | Madhya Pradesh | Tamil Nadu | Puducherry |
|---|---|---|---|---|---|---|---|---|---|---|---|---|---|---|
|  | N = 53,212 | n = 3,508 | n = 1,764 | n = 3,208 | n = 1,705 | n = 6,541 | n = 13,651 | n = 2,933 | n = 4,048 | n = 4,340 | n = 4,265 | n = 6,803 | n = 3,794 | n = 582 |
| Poorest ® | 1.00 | 1.00 | 1.00 | 1.00 | 1.00 | 1.00 | 1.00 | 1.00 | 1.00 | 1.00 | 1.00 | 1.00 | 1.00 | 1.00 |
| Poorer | 1.04 | 0.92 | 1.12 | 0.98 | 0.95 | 1.00 | 1.08 | 0.93 | 0.90 | 0.89 | 1.28 | 1.15 | 1.40 | 0.18 |
| Middle | 1.11* | 1.97 | 1.63 | 1.07 | 1.02 | 1.20 | 1.20* | 0.89 | 1.25 | 0.86 | 1.27 | 1.28* | 1.23 | 0.89 |
| Richer | 1.14** | 1.56 | 1.43 | 1.36 | 0.96 | 1.14 | 1.18 | 1.11 | 1.41 | 1.13 | 1.22 | 1.34* | 1.12 | 0.33* |
| Richest | 1.34*** | 1.84 | 0.95 | 1.46 | 1 | 1.26 | 1.38** | 1.00 | 1.12 | 1.42 | 1.44 | 1.41* | 1.14 | 1.00 |
| Residence |  |  |  |  |  |  |  |  |  |  |  |  |  |  |
| Urban ® | 1.00 | 1.00 | 1.00 | 1.00 | 1.00 | 1.00 | 1.00 | 1.00 | 1.00 | 1.00 | 1.00 | 1.00 | 1.00 | 1.00 |
| Rural | 1.17*** | 1.05 | 0.73 | 0.93 | 1.18 | 1.19 | 0.98 | 1.35 | 0.89 | 1.33* | 1.31 | 1.21* | 0.84 | 0.93 |
| Had media access |  |  |  |  |  |  |  |  |  |  |  |  |  |  |
| Yes ® | 1.00 | 1.00 | 1.00 | 1.00 | 1.00 | 1.00 | 1.00 | 1.00 | 1.00 | 1.00 | 1.00 | 1.00 | 1.00 | 1.00 |
| No | 1.01 | 1.06 | 1.03 | 1.04 | 0.92 | 0.93 | 0.82*** | 0.94 | 0.94 | 1.03 | 0.89 | 0.88 | 0.90 | 1.38 |
| Constant | 0.09*** | 0.07*** | 0.07*** | 0.07*** | 0.11*** | 0.13*** | 0.12*** | 0.07*** | 0.12*** | 0.09*** | 0.04*** | 0.13** | 0.06*** | 0.03* |
| Model fit statistics |  |  |  |  |  |  |  |  |  |  |  |  |  |  |
| F-statistic | 217.1*** | 23.5*** | 14.9*** | 23.6*** | 8.0*** | 36.3*** | 80.3*** | 9.7*** | 17.7*** | 17.1*** | 22.2*** | 43.4*** | 7.2*** | 6.4*** |
| ICC | 0.115 | 0.056 | 0.028 | 0.080 | 0.006 | 0.050 | 0.076 | 0.111 | 0.107 | 0.064 | 0.128 | 0.051 | 0.045 | 0.229 |

Source: Authors' calculation using NFHS-5, 2019/21.

Note: ICC = Intraclass correlation coefficient. The number of cases in pooled data (N) and in states' data (n) are weighted with, respectively, national level and state level weights. Supplementary S5 Table represents 95% confidence intervals and exact P-values of the estimates.

* p < 0.05

** p < 0.01

*** p < 0.001.

(ICC = 0.120), Arunachal Pradesh (ICC = 0.190), Odisha (ICC = 0.120), Chhattisgarh (ICC = 0.176), and Madhya Pradesh (ICC = 0.174) were medium, and that effects in the remaining states/UTs were small.

It is seen from Table 4 that the cluster memberships showed a medium effect on having sex in the last 30 days in pooled data (ICC = 0.115), Arunachal Pradesh (ICC = 0.111), Jharkhand (ICC = 0.107), Chhattisgarh (ICC = 0.128), and Puducherry (ICC = 0.229), and that membership showed a small effect in the remaining states/UTs.

## Discussion and conclusions

This study examined the impact of the COVID-19 pandemic and its associated measures on fertility behaviours in a lower-middle-income country–India, which is home to approximately one-fifth of the world population. Our findings show that at the aggregate level of the selected 13 states and UTs, the women interviewed in the post-lockdown period were significantly more likely to desire another birth within two years, and less likely to use contraception and have sex than those interviewed in the pre-lockdown period. As the selected clusters (furnishing the samples for analyses) that were interviewed in the pre- and post-lockdown periods were not skewed to any particular geographic direction in India where fertility was low or high, the observed changes in fertility behaviours in this study could result from the COVID-19 pandemic.

The desire for a birth within the next two years substantially increased in the pool of the selected 13 states and UTs and in four individual states in India in the post-lockdown period. In the case of individual states, the women interviewed in the post-lockdown period in Rajasthan, Arunachal Pradesh, Chhattisgarh, and Tamil Nadu were significantly more likely to desire another birth than those interviewed in the pre-lockdown period in those states. Rajasthan, Arunachal Pradesh, and Chhattisgarh had a moderate socioeconomic status, while the socioeconomic status of Tamil Nadu was rich [33]. Studies in the upper-middle and high-income countries in Europe and North America [5, 48, 49], and in a lower-middle-income country (Bangladesh) [30] found that the pandemic was negatively associated with fertility intentions. Fertility intentions in these countries with small family norms were negatively revised, probably because of the potential economic and health uncertainties related to the pandemic. In addition to those uncertainties, increased burden on parents' time resulting from the return of childcare to home for school closure could have also contributed to that negative revision of fertility intention [31]. A study in Shanghai, however, found that the COVID-19 pandemic did not affect the fertility intentions of the couples who had trust in the health measures [29]. In the areas with strong social capital and trust, fertility declines were less in the periods with unexpected surges in economic uncertainty [50]. Nevertheless, an opposite effect may also be observed because of the differences in socioeconomic settings. About 60% of working-aged (15–59 years) women are engaged in full-time housework in India [51]. The economic and health concerns that emerge from the pandemic may not apply in an Indian setting because of its widespread poverty (25% multidimensional poverty rate) and poor maternal and child health status [24, 33]. In the Indian setting, people may rather desire extra children in the leisure of lockdown and economic downturn because of having benefits from them in the future [52]. The proportion of women desiring a birth within the next two years may also increase after the onset of the pandemic for bringing their childbearing plan forward from the consideration that their husbands and relatives may be able to devote more time to rearing the child due to the lockdown.

The use of contraceptives in the pool of the selected states remarkably declined after the lockdown. Use of contraception also declined substantially in three individual states: Uttar

Pradesh, Jharkhand, and Odisha. The Socioeconomic status of Uttar Pradesh and Jharkhand was poor, while Odisha had a moderate socioeconomic status [33]. Whereas contraceptive use in Punjab, Delhi, Arunachal Pradesh, and Tamil Nadu increased significantly after the lockdown. Punjab, Delhi, and Tamil Nadu had a rich socioeconomic status, while Arunachal Pradesh had a moderate socioeconomic status [33]. A study in an upper middle-income country (Moldova) found that there was no significant association between lockdown and modern contraceptive use [5]. Lindberg et al. argued that the decline in contraceptive use may result from women's limited access to contraception and sexual and reproductive health services and their limited affordability of these services during the pandemic [28]. If these limitations occur during the pandemic, they should result in an increase in the unmet need for family planning. Our additional calculations from the pooled data showed that the difference between the unmet needs for family planning after and before the lockdown was nonsignificant (NFHS-5 raw data). This non-increasing unmet need for family planning could be due to the increase in the desire for another birth in the near future. Nevertheless, reporting of the unmet need for family planning by women in Uttar Pradesh, Jharkhand, and Odisha increased significantly after the lockdown (an additional calculation from NFHS-5 data). This increased reporting of the unmet need for family planning in those states after the lockdown might be the result of the increase in women's limitations of access to and affordability of contraception and family planning services caused by the pandemic. In low-income regions, the pandemic-driven economic downturns (such as increases in unemployment and underemployment) led to reduced purchasing power [31]. In addition, virtually nonexistence of transport to reach facilities, non-supply, lack of human resources, health workers' unwillingness to work in unsafe conditions, and loss of distribution severely limited access to family planning in India during the pandemic [19].

Sexual intercourse declined significantly in the pool of the selected states. Specifically, the women who were interviewed after the lockdown in Punjab, Uttar Pradesh, Madhya Pradesh, and Tamil Nadu were significantly less likely to have sex than those who were interviewed before the lockdown. Among these, Punjab and Tamil Nadu had rich socioeconomic status, while Uttar Pradesh and Madhya Pradesh had poor socioeconomic status [33]. A study in United Kingdom (a high-income country) also found a reduced sexual activities during the pandemic [53]. This reduced sexual activity might result from the maintenance of social distancing to prevent the transmission of COVID-19 through contact with droplets in the partner's nose, mouth, and saliva [54]. Besides this, our results show that the women in the Haryana state (one of the richest states in India) were having sex more frequently after the lockdown. The study in Moldova also found that people were having sex more frequently in the post-lockdown period compared to the pre-lockdown period [5]. People spent more time with their partners at home due to the disruption of their daily activities due to the self-isolation in the COVID-19 pandemic. Spending this extended time with partners is perhaps partially responsible for those increased sexual activities [55]. As there is no article to support this hypothesis, independent studies are required to understand this phenomenon.

Our findings show that the poverty-stricken states of Uttar Pradesh and Jharkhand, and the Odisha state with moderate socioeconomic status experienced a significant decrease in contraceptive use and non-significant changes in the desire for a birth and sexual activities after the lockdown. Therefore, changes in fertility behaviours in these states are likely to exert an increasing effect on their fertilities. On the contrary, contraceptive use significantly increased after the lockdown in the rich states of Punjab, Delhi, and Tamil Nadu, and in the Arunachal Pradesh state with moderate socioeconomic status. Sexual activities in Punjab and Tamil Nadu declined significantly after the lockdown, while changes in sexual activities in Delhi and Arunachal Pradesh were non-significant. The desire for a birth in Arunachal Pradesh and Tamil

Nadu increased significantly after the lockdown, but changes in that desire in Punjab and Delhi were non-significant after the lockdown. These changes in fertility behaviours in Punjab, Delhi, Tamil Nadu, and Arunachal Pradesh may exert a reducing effect on their fertilities. In the remaining study states, the effects of the changes in fertility behaviours may remain neutral. The fertility-changing effects of the changes in the fertility behaviours in the study states and UTs could be further adjusted by the trend in the prevalence of abortion.

One of the main strengths of this study is that it has used advanced analysis technique that takes complex sample design effects into account. Findings of this study will add new knowledge on the association between the COVID-19 pandemic and fertility behaviour in a lower-middle-income country with a very large population and land area. This study is based on a large nationally representative sample that were collected using a sophisticated sampling technique and therefore the findings of this study are expected to be more reliable and conclusive.

It is also important to explore the limitations of this study. Because of the unavailability of data, this study could not examine changes in the concurrent steps taken to give birth. Nevertheless, the analysis of the desire for another birth in this study gave an idea about the influence of the pandemic on women's interest in having a child in two years [56]. Our study also could not analyze the influence of having a COVID-19 infection on changes in fertility behaviours because data were unavailable. Our study intended to investigate the influence of the pandemic on fertility behaviours not only among those infected with COVID-19 but also among others in the community. Our independent variable–interview period–captured the influence of the pandemic among all people irrespective of their COVID-19 infection status therefore that independent variable served well in achieving the intention of this study. Abortion played an important role in the fertility transition in India [57]. Nonetheless, our study could not examine the influence of the pandemic on abortion incidence because data were unavailable. Our additional calculations using raw data from the NFHS-5 (2019/21) showed that the difference between desires for a birth within two years in India in the pre-lockdown (52.7%, 95% CI: 52.1–53.3) and post-lockdown (52.4%, 95% CI: 51.7–53.2) periods was nonsignificant. This nonsignificant change in the desire for a birth and the disruption in health care and family planning services in India because of the pandemic were unfavorable to the increase in the prevalence of abortion [58]. As this is a cross-sectional study, it only provides information on the statistical association between the variables in question. Independent studies in the states that experienced significant changes in fertility behaviours could reveal more detailed information on the factors associated with those changes.

Despite the limitations, this study has provided important insights into the changes in certain fertility behaviours in the period following the COVID-19 lockdown in a lower middle-income country (India) in South Asia. This study shows that the impact of COVID-19 on fertility behaviours in a large part of India was different from that in other middle- and lower-middle-income countries (Moldova and Bangladesh). Contrary to the findings in Moldova and Bangladesh, the selected Indian states at the aggregate level after the lockdown experienced a significant increase in the proportion of women desiring another child and a significant decrease in contraceptive use. These changes in desire for a child and contraceptive use may exert an increasing effect on fertility at the aggregate level in the selected states. Nevertheless, this fertility-increasing effect can be compensated by reduced sexual activities. The pandemic, however, showed mixed effects on fertility behaviours at the state level. Our findings indicate that among the included states, changes in fertility behaviours after the COVID-19 lockdown in the poverty-stricken states may exert either a positive or neutral effect on their fertility, and that changes in the rich states may exert either a reducing or neutral effect on their fertility, while the effects of fertility behavioural changes in the states with moderate socioeconomic status could be negative, positive, or neutral in the near future. The aggregate

effect of the behavioural changes on fertility could be further adjusted by the trend in abortion incidence. The states that experienced pro-natalist changes in fertility behaviours (Uttar Pradesh, Jharkhand, and Odisha) constitute 23.6% of the Indian population, which is much higher than that of those that experienced anti-natalist changes in fertility behaviours (Punjab, Delhi, Tamil Nadu, and Arunachal Pradesh) (9.4%) [34]. Therefore, the influence of the fertility trends in the states with pro-natalist changes on the whole India's recent fertility trend could be greater than that of those with anti-natalist changes in fertility behaviours. Changes in fertility behaviours in the Indian states that were analysed in this study could be reflected in their fertility estimates in the couple of years following the NFHS-5 survey. Therefore, we have to wait until the release of reliable data to understand the effects of those changes in fertility behaviours on fertility in India and its states.

## Supporting information

**S1 Checklist. STROBE statement—Checklist of items that should be included in reports of** *cross-sectional studies.*
(DOC)

**S2 Checklist.** *PLOS ONE* **clinical studies checklist.**
(DOCX)

**S1 Table. Selected weighted cases for each dependent variable interviewed in pre- and post-lockdown periods in the selected Indian states and UTs, 2019/21.**
(DOCX)

**S2 Table. Percentage of the pooled women selected for each analysis by interview period and selected characteristics, India 2019/21.**
(DOCX)

**S3 Table. 95% confidence intervals (CI) and P-values of the adjusted odds ratios of desiring a birth within 2 years among the women interviewed in pre- and post-lockdown periods in the selected Indian states, 2019/21.**
(DOCX)

**S4 Table. 95% confidence intervals (CI) and P-values of the adjusted odds ratios of using contraception among the women interviewed in pre- and post-lockdown periods in the selected Indian states, 2019/21.**
(DOCX)

**S5 Table. 95% confidence intervals (CI) and P-values of the adjusted odds ratios of having sex in the last 30 days among the women interviewed in pre- and post-lockdown periods in the selected Indian states, 2019/21.**
(DOCX)

## Acknowledgments

The authors are thankful to the Demographic and Health Survey Program for providing permission of using the National Family Health Survey-5 data of India.

## Author Contributions

**Conceptualization:** Md. Mahfuzur Rahman, Manas Ranjan Pradhan.

**Data curation:** Md. Mahfuzur Rahman, Manoj Kumer Ghosh, Md. Moshfiqur Rahman.

**Formal analysis:** Md. Mahfuzur Rahman.

**Methodology:** Md. Mahfuzur Rahman.

**Software:** Md. Mahfuzur Rahman.

**Validation:** Manas Ranjan Pradhan.

**Visualization:** Md. Mahfuzur Rahman, Manoj Kumer Ghosh.

**Writing – original draft:** Md. Mahfuzur Rahman, Manas Ranjan Pradhan.

**Writing – review & editing:** Manas Ranjan Pradhan, Md. Moshfiqur Rahman.

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
