## [Decision Letter · Decision Letter 0]

18 Jul 2024

PONE-D-23-42397The impact of COVID-19 on fertility behaviour in Indian states: Evidence from the National Family Health Survey (2019/21)PLOS ONE

Dear Dr. Mahfuzur,

Thank you for submitting your manuscript to PLOS ONE. After careful consideration, we feel that it has merit but does not fully meet PLOS ONE’s publication criteria as it currently stands. Therefore, we invite you to submit a revised version of the manuscript that addresses the points raised during the review process.

We look forward to receiving your revised manuscript.

Kind regards,

Ashish Wasudeo Khobragade, MD

Academic Editor

PLOS ONE

2. For studies involving third-party data, we encourage authors to share any data specific to their analyses that they can legally distribute. PLOS recognizes, however, that authors may be using third-party data they do not have the rights to share. When third-party data cannot be publicly shared, authors must provide all information necessary for interested researchers to apply to gain access to the data. (https://journals.plos.org/plosone/s/data-availability#loc-acceptable-data-access-restrictions)

a) A description of the data set and the third-party source

b) If applicable, verification of permission to use the data set

c) Confirmation of whether the authors received any special privileges in accessing the data that other researchers would not have

d) All necessary contact information others would need to apply to gain access to the data

3. We note that Figures 2 and 4 in your submission contain [map/satellite] images which may be copyrighted. All PLOS content is published under the Creative Commons Attribution License (CC BY 4.0), which means that the manuscript, images, and Supporting Information files will be freely available online, and any third party is permitted to access, download, copy, distribute, and use these materials in any way, even commercially, with proper attribution. For these reasons, we cannot publish previously copyrighted maps or satellite images created using proprietary data, such as Google software (Google Maps, Street View, and Earth). For more information, see our copyright guidelines: http://journals.plos.org/plosone/s/licenses-and-copyright.

a. You may seek permission from the original copyright holder of Figures 2 and 4 to publish the content specifically under the CC BY 4.0 license. 

Additional Editor Comments:

1. Provide a structured abstract.

2. The 95% CI of odds ratios and exact P values are not mentioned in Tables 2, 3 and 4 of the adjusted odds ratios. Please, organize the tables properly.

3. The mixed logit model was used in the study. Adjusted odds ratios are mentioned. However, other statistical details of the model for the pooled data are lacking in the manuscript, e.g., the goodness of fit test, Bayesian information criterion etc.

Reviewers' comments:

Reviewer's Responses to Questions

**Comments to the Author**

1. Is the manuscript technically sound, and do the data support the conclusions?

Reviewer #1: Yes

Reviewer #2: Yes

2. Has the statistical analysis been performed appropriately and rigorously? 

Reviewer #1: Yes

Reviewer #2: Yes

3. Have the authors made all data underlying the findings in their manuscript fully available?

Reviewer #1: Yes

Reviewer #2: No

4. Is the manuscript presented in an intelligible fashion and written in standard English?

Reviewer #1: Yes

Reviewer #2: Yes

5. Review Comments to the Author

Reviewer #1: The manuscript is technically sound and the conclusions derived are well supported by the available data and statistical analysis performed.

Authors have also performed appropriate statistical analysis for the available data to derive at the results and conclusions from the study.

Authors have also made accessible relevant data to support the findings of the study.

The manuscript was also presented in clear and well communicated English.

The reviewer however will want the authors to address the following

*Replace "this" with "the", in 'this percentage for those selected for analyzing', line 302.

*By what measure was systematic excluded? , in the statement, 'No remarkable systematic bias was observed in the proportions of women with different characteristics interviewed in the pre- and post lockdown periods.'. Lines 307-309

* Re format Table 2, line 346. The figures in the columns were mal aligned with the rows titled, "Religion, Wealth status, Residence and Media access"

Reviewer #2: Thank you for your effort in this study,:

I think you only assess the effect of sociodemographic characters on the fertility behavior, so I think that must be presented in the title of the research .

This drive us to the question why you didn't assess other factors related to health status or infection with covid?? I think these factors are important and gave role in the fertility motives of the population

Please make the limitation of the study and recomendations more clear

6. PLOS authors have the option to publish the peer review history of their article (what does this mean?). If published, this will include your full peer review and any attached files.

Reviewer #1: No

Reviewer #2: No

---

## [Author Response · Author response to Decision Letter 0]

30 Aug 2024

Point-wise response to editor’s and reviewers’ comments

Note: All the line numbers refer to the file named “Revised Manuscript with Track Changes”.

Authors’ Response

Formatted as per instructions.

2. For studies involving third-party data, we encourage authors to share any data specific to their analyses that they can legally distribute. PLOS recognizes, however, that authors may be using third-party data they do not have the rights to share. When third-party data cannot be publicly shared, authors must provide all information necessary for interested researchers to apply to gain access to the data. (https://journals.plos.org/plosone/s/data-availability#loc-acceptable-data-access-restrictions)

a) A description of the data set and the third-party source

b) If applicable, verification of permission to use the data set

c) Confirmation of whether the authors received any special privileges in accessing the data that other researchers would not have

d) All necessary contact information others would need to apply to gain access to the data

Authors’ Response

All the line numbers refer to the file named “Revised Manuscript with Track Changes”. We have added notes on the availability of data in the data subsection under the methodology section in the manuscript and manuscript submission process. For seeing the notes in the manuscript, please see the lines 208–210.

3. We note that Figures 2 and 4 in your submission contain [map/satellite] images which may be copyrighted. All PLOS content is published under the Creative Commons Attribution License (CC BY 4.0), which means that the manuscript, images, and Supporting Information files will be freely available online, and any third party is permitted to access, download, copy, distribute, and use these materials in any way, even commercially, with proper attribution. For these reasons, we cannot publish previously copyrighted maps or satellite images created using proprietary data, such as Google software (Google Maps, Street View, and Earth). For more information, see our copyright guidelines: http://journals.plos.org/plosone/s/licenses-and-copyright.

a. You may seek permission from the original copyright holder of Figures 2 and 4 to publish the content specifically under the CC BY 4.0 license. 

Authors’ Response

The maps (Figures 2 & 4) used in our article were not copyrighted. We generated Fig 2 using the shape file from the database of Runfold et al. (Runfola D, Anderson A, Baier H, Crittenden M, Dowker E, Fuhrig S, et al. geoBoundaries: A global database of political administrative boundaries. PLoS ONE. 2020; 15(4):e0231866.). We have clarified the note with the title of the Fig 2 (please see the title and note of Fig 2). The Fig 4 was produced using the shape file provided by the 2019/21 National Family and Health Survey (available on https://dhsprogram.com/ subject to an online request) and the database of Runfold et al. (we have clarified these through the notes in Fig 4). Both figures were produced using ArcGIS desktop version. We have explained these in lines 309–315. The https://dhsprogram.com/ granted us access to the shape file with due process, and the database of Runfold et al. is freely available to everyone. Therefore, we don’t need to obtain any permission for using those maps, and there is no restriction in using and distributing the figures.

Authors’ Response

We have newly added following references:

41. Murphy KR, Myors B. Statistical power analysis: A simple and general model for traditional and modern hypothesis tests. Mahwah, NJ: Lawrence Erlbaum [Internet]. 1998 [cited 2024 Aug 17]. Available from: https://www.routledge.com/Statistical-Power-Analysis-A-Simple-and-General-Model-for-Traditional-and-Modern-Hypothesis-Tests-Fifth-Edition/Myors-Murphy/p/book/9781032283005

42. LeBreton JM, Senter JL. Answers to 20 questions about interrater reliability and interrater agreement. Organ Res Methods. 2008; 11(4):815–52.

43. StataCorp. Stata: Release 16. Statistical Software. College Station, TX: StataCorp LLC [Internet]. 2019 [cited 2020 Apr 20]. Available from: https://www.stata.com/manuals/me.pdf

Besides these we have checked the correctness and completeness of the references.

Authors’ overall Response

The authors are thankful to the editor and reviewers for reviewing this article and providing constructive suggestions for improving our article. We have effected all the changes suggested by the editor and reviewers with due explanation. Besides the suggestions of the reviewers, we tried to improve our article by clarifying sentences and correcting typographic mistakes. A list of such corrections are as follows (All the line numbers refer to the file named “Revised Manuscript with Track Changes”.):

i) We have included the word pandemic in the title to make that more appropriate. Please see the line 1 

ii) In Fig 3, in a box we wrote “Case selection for analyzing modern contraceptive use”, but that should be “Case selection for analysing contraceptive use”, and we have corrected accordingly.

iii) We have revised following lines to improve the sentences: l27–l28, l35, l36, l42, l43, l44, l49–l57, l349, l375–l377, l389–l391, l403–l405, l510–512, l522–524, l556, l561

iv) We have corrected some typographic mistakes in the tables as well, which are as follows: subcategories under current age and caste in Table 2, 3 and 4, Odds ratios for religion in Tamil Nadu and Puducherry in Table 2, Odds ratios for wealth status in Delhi in Table 2, Odds ratio for level of schooling in Puducherry in Table 4, Odds ratio for wealth status in Delhi and Puducherry in Table 4, replacing modern contraceptive use with contraceptive use in S1 Table, and including data for number of children and had media access in S2 Table, 

v) We have corrected typographic mistakes in lines l250, l257, l262, l284, l288, l289, l324–l328.

Additional Editor Comments:

1. Provide a structured abstract.

Authors’ Response

We have made a structured abstract.

2. The 95% CI of odds ratios and exact P values are not mentioned in Tables 2, 3 and 4 of the adjusted odds ratios. Please, organize the tables properly.

Authors’ Response

Thanks for this suggestion, because CIs and P values are very important. We have provided CIs and P values in supplementary tables because of the following reasons. We have fitted 14 models for each dependent variable; as a result, our tables containing the results of multivariate analysis (Table 2, 3, and 4) already have become very wide. If we put CI and P value in addition to odds ratio in the same table, each of the table will become extremely wide, which will reduce the readability of the tables and paper. Considering all these facts, we have put the CIs and P values for Table 2, 3, and 4 in the supplementary S3, S4, and S5 Tables, respectively. You can find those supplementary tables in supplementary file “S1, S2, S3, S4, and S5 Tables”.

3. The mixed logit model was used in the study. Adjusted odds ratios are mentioned. However, other statistical details of the model for the pooled data are lacking in the manuscript, e.g., the goodness of fit test, Bayesian information criterion etc.

Authors’ Response

In revision, we have reanalyzed our data using Stata 17.0 (the initial analyses were done with the Stata 15.0). The Stata 17.0 can produce intraclass correlation coefficients (ICC). We have put ICC and result of F-test for each model (Please see the last two rows of the Tables 2, 3, and 4). F-statistic and ICC could be considered goodness of fit. We have explained about F-statistic and ICC in the statistical analysis section (please see the lines 297–302). We have also explained these in the result section in lines 407–421. We employed mixed-effects logit regression analysis with complex sampling design, which did not produce Bayesian Information Criterion (BIC) and Akaike Information Criterion (AIC). BIC and AIC are usually used for model comparison. In our study, we employed logit model because our variables were categorical. As we did not fit any model other than logit, we did not have any model to compare with using BIC or AIC. 

Reviewers' comments:

Reviewer's Responses to Questions

Comments to the Author

1. Is the manuscript technically sound, and do the data support the conclusions?

Reviewer #1: Yes

Reviewer #2: Yes

2. Has the statistical analysis been performed appropriately and rigorously?

Reviewer #1: Yes

Reviewer #2: Yes

3. Have the authors made all data underlying the findings in their manuscript fully available?

Reviewer #1: Yes

Reviewer #2: No

4. Is the manuscript presented in an intelligible fashion and written in standard English?

Reviewer #1: Yes

Reviewer #2: Yes

5. Review Comments to the Author

Reviewer #1: The manuscript is technically sound and the conclusions derived are well supported by the available data and statistical analysis performed.

Authors have also performed appropriate statistical analysis for the available data to derive at the results and conclusions from the study.

Authors have also made accessible relevant data to support the findings of the study.

The manuscript was also presented in clear and well communicated English.

The reviewer however will want the authors to address the following

*Replace "this" with "the", in 'this percentage for those selected for analyzing', line 302.

Authors’ Response

Thank you very much for reviewing our paper. We have effected the change as per your suggestion. Please see the line 327 (All the line numbers refer to the file named “Revised Manuscript with Track Changes”.).

*By what measure was systematic excluded? , in the statement, 'No remarkable systematic bias was observed in the proportions of women with different characteristics interviewed in the pre- and post lockdown periods.'. Lines 307-309

Authors’ Response

Thanks for identifying this confusing sentence. We have corrected the sentence. Please see the lines 332–334.

* Re format Table 2, line 346. The figures in the columns were mal aligned with the rows titled, "Religion, Wealth status, Residence and Media access"

Authors’ Response

These corrections have been done. Please see the rows of Religion, Wealth status, Residence and Media access in Table 2. Besides these, we have examined other Tables as well, and have revised the inconsistent alignments and typographic mistakes.

Reviewer #2: Thank you for your effort in this study,:

I think you only assess the effect of sociodemographic characters on the fertility behavior, so I think that must be presented in the title of the research .

This drive us to the question why you didn't assess other factors related to health status or infection with covid?? I think these factors are important and gave role in the fertility motives of the population

Authors’ Response

Thank you very much for reviewing our paper and for your valuable c

---

## [Decision Letter · Decision Letter 1]

3 Oct 2024

PONE-D-23-42397R1The impact of COVID-19 pandemic on fertility behaviour in Indian states: Evidence from the National Family Health Survey (2019/21)PLOS ONE

Dear Dr. Mahfuzur,

Thank you for submitting your manuscript to PLOS ONE. After careful consideration, we feel that it has merit but does not fully meet PLOS ONE’s publication criteria as it currently stands. Therefore, we invite you to submit a revised version of the manuscript that addresses the points raised during the review process.

We look forward to receiving your revised manuscript.

Kind regards,

Ashish Wasudeo Khobragade, MD

Academic Editor

PLOS ONE

Journal Requirements:

Reviewers' comments:

Reviewer's Responses to Questions

**Comments to the Author**

1. If the authors have adequately addressed your comments raised in a previous round of review and you feel that this manuscript is now acceptable for publication, you may indicate that here to bypass the “Comments to the Author” section, enter your conflict of interest statement in the “Confidential to Editor” section, and submit your "Accept" recommendation.

Reviewer #1: All comments have been addressed

Reviewer #2: All comments have been addressed

Reviewer #3: (No Response)

2. Is the manuscript technically sound, and do the data support the conclusions?

Reviewer #1: (No Response)

Reviewer #2: No

Reviewer #3: Partly

3. Has the statistical analysis been performed appropriately and rigorously? 

Reviewer #1: (No Response)

Reviewer #2: Yes

Reviewer #3: No

4. Have the authors made all data underlying the findings in their manuscript fully available?

Reviewer #1: (No Response)

Reviewer #2: Yes

Reviewer #3: No

5. Is the manuscript presented in an intelligible fashion and written in standard English?

Reviewer #1: (No Response)

Reviewer #2: Yes

Reviewer #3: No

6. Review Comments to the Author

Reviewer #1: (No Response)

Reviewer #2: Thank you for your effort in addressing all my comments.

I hope you to take the comments in consideration in your future researches

I wish you good luck

Reviewer #3: This paper addresses a valuable topic and offers some great insights into the impact of COVID-19 on fertility behaviors. That said, there are a few areas that could be improved. Providing more background information would help set the stage for readers from other cultural backgrounds, and improving the overall flow of the argument could make it easier to follow. Also, expanding the statistical analysis section with more details would be helpful for clarity and reproducibility. With these adjustments, the paper could be even stronger.

1. It would be helpful to give more background on India’s fertility trends for readers who might not be familiar with the country’s unique situation. E.g., explain a bit more about why the mean ideal family size and fertility rate have been dropping over the years?

2. Make necessary connections/transitions in the introduction section. E.g., “Since the pandemic disrupted these trends, it might be worth expanding on how exactly COVID-19 impacted fertility behaviors.”

3. The terms "direction" and "tempo" of fertility change are important but may benefit from further explanation, particularly for an international audience. A deeper explanation of these concepts in the context of pandemic-related disruptions would add depth to the discussion and broaden the accessibility of the paper for readers unfamiliar with India's demographic landscape.

4. Line 101: “A study showed that fertility intentions in the United Kingdom, …” – add citation of this study.

5. In Socioeconomic profiles section, you provided MPR for all 13 states. Where does these numbers come from? Proper reference is needed to ensure robustness.

6. Accurate data is crucial for producing robust results. Figure 3 are very blurry and not readable so I’m not sure whether the following question have been addressed there: What measures have been put in place to ensure that respondents provide truthful information on sensitive topics like their desire for birth within the next two years, current contraceptive use, and recent sexual activity? This information is highly personal.

7. All figures are very blurry and not legible. Higher resolution figures are needed.

8. Line 245: what’s the definition of “clusters” here?

9. Line 329: “Among the women interviewed in the pre- and post-lockdown periods, more than 80% were Hindu” – elaborate more on how this relates to the analysis. Are Hindus more comfortable to be interviewed compared with others? More context/background information are needed here.

10. The Statistical Analysis section should clearly explain why the two-level mixed effects logit model was used for multivariate analysis. E.g. The data in this study is organized hierarchically: Individuals (level 1) are nested within clusters (level 2) - use two-level mixed effects logit model to address this structure; Capturing Cluster-Level Variation, etc. Current description is not clearly for repeating the analysis.

11. Tables: provide CI calculation method as table footnotes.

12. Why were the 13 states classified based on socioeconomic status? Several conclusions are mentioned in the discussion section. For example, in states with poorer socioeconomic conditions (such as Uttar Pradesh, Jharkhand, and Odisha), contraceptive use significantly declined after the lockdown. It is necessary to further explain the possible reasons for these phenomena, in order to better understand why differences in socioeconomic conditions lead to such disparities.

7. PLOS authors have the option to publish the peer review history of their article (what does this mean?). If published, this will include your full peer review and any attached files.

Reviewer #1: No

Reviewer #2: No

Reviewer #3: **Yes: **Yuhang Liu

---

## [Author Response · Author response to Decision Letter 1]

30 Oct 2024

PONE-D-23-42397R1

The impact of COVID-19 pandemic on fertility behaviour in Indian states: Evidence from the National Family Health Survey (2019/21)

PLOS ONE

Dear Dr. Mahfuzur,

Thank you for submitting your manuscript to PLOS ONE. After careful consideration, we feel that it has merit but does not fully meet PLOS ONE’s publication criteria as it currently stands. Therefore, we invite you to submit a revised version of the manuscript that addresses the points raised during the review process.

We look forward to receiving your revised manuscript.

Kind regards,

Ashish Wasudeo Khobragade, MD

Academic Editor

PLOS ONE

Journal Requirements:

Authors’ Response

The reference list has been checked, which is found complete and correct. We have included following additional references:

26. Gandotra, M. M., Robert D. Retherford, Arvind Pandey, Norman Y. Luther, and Vinod Mishra. 1998. Fertility in India. 9. Mumbai, India and Honolulu, Hawaii, U.S.A.: International Institute for Population Sciences and East-West Center Program on Population.

27. Bhat, P. N. Mari, and A. J. Francis Zavier. 2003. “Fertility Decline and Gender Bias in Northern India.” Demography 40(4):637–57. doi: 10.2307/1515201.

38. International Institute for Population Sciences (IIPS). National Family Health Survey 2019-20 NFHS-5: Data quality assurance and quality control mechanisms 2019-20 [Internet]. Mumbai, India; 2019 [cited 2024 Oct 18]. Available from: https://www.nfhsiips.in/nfhsuser/assets/pdf/NFHS%20data%20quality%20assurance.pdf

39. Singh SK, Lhungdim H, Shekhar C, Dwivedi LK, Pedgaonkar S, James KS. Innovative field procedures in a large-scale survey to ensure quality of data in pandemic situation: Evidence from NFHS-5, 2019-21. Demogr India. 2022; 51(1):17–39.

48. Kramer S. Religious composition of India. Pew Research Center [Internet]. 2021 [cited 2024 Oct 19]. Available from: https://www.pewresearch.org/wp-content/uploads/sites/20/2021/09/PF_09.21.21_Religious-Composition-of-India-FULL.pdf

51. Aassve A, Le Moglie M, Mencarini L. Trust and fertility in uncertain times. Popul Stud. 2021; 75(1):19–36.

55. Masoudi M, Maasoumi R, Bragazzi NL. Effects of the COVID-19 pandemic on sexual functioning and activity: a systematic review and meta-analysis. BMC Public Health. 2022; 22(1):189.

56. Jacob L, Smith L, Butler L, Barnett Y, Grabovac I, McDermott D, et al. Challenges in the practice of sexual medicine in the time of COVID-19 in the United Kingdom. J Sex Med. 2020; 17(7):1229.

Reviewers' comments:

Reviewer's Responses to Questions

Comments to the Author

1. If the authors have adequately addressed your comments raised in a previous round of review and you feel that this manuscript is now acceptable for publication, you may indicate that here to bypass the “Comments to the Author” section, enter your conflict of interest statement in the “Confidential to Editor” section, and submit your "Accept" recommendation.

Reviewer #1: All comments have been addressed

Reviewer #2: All comments have been addressed

Reviewer #3: (No Response)

2. Is the manuscript technically sound, and do the data support the conclusions?

Reviewer #1: (No Response)

Reviewer #2: No

Reviewer #3: Partly

3. Has the statistical analysis been performed appropriately and rigorously?

Reviewer #1: (No Response)

Reviewer #2: Yes

Reviewer #3: No

4. Have the authors made all data underlying the findings in their manuscript fully available?

Reviewer #1: (No Response)

Reviewer #2: Yes

Reviewer #3: No

5. Is the manuscript presented in an intelligible fashion and written in standard English?

Reviewer #1: (No Response)

Reviewer #2: Yes

Reviewer #3: No

6. Review Comments to the Author

Reviewer #1: (No Response)

Reviewer #2: Thank you for your effort in addressing all my comments.

I hope you to take the comments in consideration in your future researches

I wish you good luck

Reviewer #3: This paper addresses a valuable topic and offers some great insights into the impact of COVID-19 on fertility behaviors. That said, there are a few areas that could be improved. Providing more background information would help set the stage for readers from other cultural backgrounds, and improving the overall flow of the argument could make it easier to follow. Also, expanding the statistical analysis section with more details would be helpful for clarity and reproducibility. With these adjustments, the paper could be even stronger.

Authors’ Response

The authors would like to thank the reviewer for his contribution to improve our paper. We have addressed all the issues showed by the reviewer and respond the questions of the reviewer with due consideration. We have thoroughly revised the paper and tried to improve the clarification beyond the comment of the reviewer (for example ls 269–276). ls means lines and all the line numbers refer to the lines in the file named “Revised Manuscript with Track Changes”

1. It would be helpful to give more background on India’s fertility trends for readers who might not be familiar with the country’s unique situation. E.g., explain a bit more about why the mean ideal family size and fertility rate have been dropping over the years?

Authors’ Response

We have added an explanation on the aforementioned matter in the ls 92–102.

2. Make necessary connections/transitions in the introduction section. E.g., “Since the pandemic disrupted these trends, it might be worth expanding on how exactly COVID-19 impacted fertility behaviors.”

Authors’ Response

This suggestion has been effected in the ls 103–108.

3. The terms "direction" and "tempo" of fertility change are important but may benefit from further explanation, particularly for an international audience. A deeper explanation of these concepts in the context of pandemic-related disruptions would add depth to the discussion and broaden the accessibility of the paper for readers unfamiliar with India's demographic landscape.

Authors’ Response

The terms "direction" and "tempo" of fertility were vague in the context. We have replaced those terms by an appropriate word that we actually meant. Please see the ls 103–106.

4. Line 101: “A study showed that fertility intentions in the United Kingdom, …” – add citation of this study.

Authors’ Response

Citation added. Please see the line 115

5. In Socioeconomic profiles section, you provided MPR for all 13 states. Where does these numbers come from? Proper reference is needed to ensure robustness.

Authors’ Response

We have made the citation clear in the ls 181–187.

6. Accurate data is crucial for producing robust results. Figure 3 are very blurry and not readable so I’m not sure whether the following question have been addressed there: What measures have been put in place to ensure that respondents provide truthful information on sensitive topics like their desire for birth within the next two years, current contraceptive use, and recent sexual activity? This information is highly personal.

Authors’ Response

The figures we provided were very clear, but unfortunately figures got blurred when the complete manuscript file was reproduced by the PLOSONE submission tool. So, we think here the PLOSONE submission tool has scope of work.

NFHS follows ethical protocols at every level of data collection and processing, including informed consent and voluntary participation, ensuring the privacy and confidentiality of data. The field investigators went through rigorous training on survey tools and interview techniques, including gathering sensitive personal information (For example, desire for birth within the next two years, current contraceptive use, and recent sexual activity). NFHS further adheres to the protocol of maintaining a gender-sensitive team composition with a mix of an adequate number of male and female investigators who belong to the native state and have proficiency in the local language(s). Further, a multi-layer monitoring of fieldwork is adopted to strengthen the data quality, including spot checks, back checks, review of field check tables and continuous supportive supervision. Field staff from IIPS conducted spot and back checks of surveyed households in a minimum of 10% of PSUs that were randomly selected by the IIPS central office. The questions used in the NFHSs are integral parts of DHS modules and are asked to women in around 90 countries where DHS surveys are being conducted 

(References 

1. International Institute for Population Sciences (IIPS). 2019. National Family Health Survey 2019-20 NFHS-5: Data Quality Assurance and Quality Control Mechanisms 2019-20. Mumbai, India.

1. Singh, S. K., H. Lhungdim, Chander Shekhar, L. K. Dwivedi, S. Pedgaonkar, and K. S. James. 2022. “Innovative Field Procedures in a Large-Scale Survey to Ensure Quality of Data in Pandemic Situation: Evidence from NFHS-5, 2019-21.” Demography India 51(1):17–39.)

We have collated some important information regarding the accuracy of data in the manuscript in the ls 240–252.

7. All figures are very blurry and not legible. Higher resolution figures are needed.

Authors’ Response

The figures we provided were very clear, but unfortunately figures got blurred when the complete manuscript file was reproduced by the PLOSONE submission tool. So, we think here the PLOSONE submission tool has scope of work.

8. Line 245: what’s the definition of “clusters” here?

Authors’ Response

Definition of cluster was provided in the ls 238–239.

9. Line 329: “Among the women interviewed in the pre- and post-lockdown periods, more than 80% were Hindu” – elaborate more on how this relates to the analysis. Are Hindus more comfortable to be interviewed compared with others? More context/background information are needed here.

Authors’ Response

Actually, that section briefly represents the group sizes among which the analysis was performed. Such high percentage of Hindu respondents was because of the fact that around 79% of people in India are Hindu (ref: Kramer, Stephanie. 2021. Religious Composition of India. Pew Research Center.). We have put a note on this fact in the ls 367–368.

10. The Statistical Analysis section should clearly explain why the two-level mixed effects logit model was used for multivariate analysis. E.g. The data in this study is organized hierarchically: Individuals (level 1) are nested within clusters (level 2) - use two-level mixed effects logit model to address this structure; Capturing Cluster-Level Variation, etc. Current description is not clearly for repeating the analysis.

Authors’ Response

We have made the description clear in the ls 327–332.

11. Tables: provide CI calculation method as table footnotes.

Authors’ Response

We have provided suggested footnotes. Please see Table 1 and supplementary tables S3, S4, and S5 Tables. 

12. Why were the 13 states classified based on socioeconomic status? Several conclusions are mentioned in the discussion section. For example, in states with poorer socioeconomic conditions (such as Uttar Pradesh, Jharkhand, and Odisha), contraceptive use significantly declined after the lockdown. It is necessary to further explain the possible reasons for these phenomena, in order to better understand why differences in socioeconomic conditions lead to such disparities.

Authors’ Response

We added the ls 178–180 to make it clear why we discussed and classified the states and union territories according to socioeconomic status.

We have revised the explanations of different conclusions (that we made) to make those explanations clearer. Our revisions regrading this include ls 477–487, ls 518–523, and ls 530–540.

7. PLOS authors have the option to publish the peer review history of their article (what does this mean?). If published, this will include your full peer review and any attached files.

Authors’ Response

Authors consent for publishing the peer review history.

Do you want your identity to be public for this peer review? For information about this choice, including consent withdra

---

## [Decision Letter · Decision Letter 2]

18 Nov 2024

The impact of COVID-19 pandemic on fertility behaviour in Indian states: Evidence from the National Family Health Survey (2019/21)

PONE-D-23-42397R2

Dear Dr. Mahfuzur,

We’re pleased to inform you that your manuscript has been judged scientifically suitable for publication and will be formally accepted for publication once it meets all outstanding technical requirements.

Kind regards,

Ashish Wasudeo Khobragade, MD

Academic Editor

PLOS ONE

Additional Editor Comments (optional):

Reviewers' comments:

Reviewer's Responses to Questions

**Comments to the Author**

1. If the authors have adequately addressed your comments raised in a previous round of review and you feel that this manuscript is now acceptable for publication, you may indicate that here to bypass the “Comments to the Author” section, enter your conflict of interest statement in the “Confidential to Editor” section, and submit your "Accept" recommendation.

Reviewer #3: All comments have been addressed

2. Is the manuscript technically sound, and do the data support the conclusions?

Reviewer #3: Yes

3. Has the statistical analysis been performed appropriately and rigorously? 

Reviewer #3: Yes

4. Have the authors made all data underlying the findings in their manuscript fully available?

Reviewer #3: Yes

5. Is the manuscript presented in an intelligible fashion and written in standard English?

Reviewer #3: Yes

6. Review Comments to the Author

Reviewer #3: My comments from the previous round has been addressed and I have no further comments. I wish you best of luck for all their future research endeavors.

7. PLOS authors have the option to publish the peer review history of their article (what does this mean?). If published, this will include your full peer review and any attached files.

Reviewer #3: **Yes: **Yuhang Liu

---

## [Editor Report · Acceptance letter]

27 Nov 2024

PONE-D-23-42397R2 

PLOS ONE

Dear Dr. Rahman, 

I'm pleased to inform you that your manuscript has been deemed suitable for publication in PLOS ONE. Congratulations! Your manuscript is now being handed over to our production team.

Kind regards, 

on behalf of

Dr. Ashish Wasudeo Khobragade 

Academic Editor

PLOS ONE